# The genetic and dietary landscape of the muscle insulin signalling network

Julian van Gerwen[1], Stewart WC Masson[1], Harry B Cutler[1], Alexis Diaz Vegas[1], Meg Potter[1], Jacqueline Stöckli[1], Søren Madsen[1], Marin E Nelson[1], Sean J Humphrey[1]*, David E James[1,2]*

[1]Charles Perkins Centre, School of Life and Environmental Sciences, University of Sydney, Sydney, Australia; [2]Faculty of Medicine and Health, University of Sydney, Sydney, Australia

*For correspondence:
sean.humphrey@mcri.edu.au
(SJH);
david.james@sydney.edu.au
(DEJ)

## Abstract
Metabolic disease is caused by a combination of genetic and environmental factors, yet few studies have examined how these factors influence signal transduction, a key mediator of metabolism. Using mass spectrometry-based phosphoproteomics, we quantified 23,126 phosphosites in skeletal muscle of five genetically distinct mouse strains in two dietary environments, with and without acute in vivo insulin stimulation. Almost half of the insulin-regulated phosphoproteome was modified by genetic background on an ordinary diet, and high-fat high-sugar feeding affected insulin signalling in a strain-dependent manner. Our data revealed coregulated subnetworks within the insulin signalling pathway, expanding our understanding of the pathway's organisation. Furthermore, associating diverse signalling responses with insulin-stimulated glucose uptake uncovered regulators of muscle insulin responsiveness, including the regulatory phosphosite S469 on Pfkfb2, a key activator of glycolysis. Finally, we confirmed the role of glycolysis in modulating insulin action in insulin resistance. Our results underscore the significance of genetics in shaping global signalling responses and their adaptability to environmental changes, emphasising the utility of studying biological diversity with phosphoproteomics to discover key regulatory mechanisms of complex traits.

## eLife assessment

This **fundamental** study provides a unique tool for assessing the range of phosphorylation in insulin reactions due to genetic variation and dietary influence through the utilization of genetically distinct mouse strains. The discoveries of this study hold substantial importance, as they shed light on the interplay between genetic attributes and environmental conditions in shaping the insulin-signaling network within skeletal muscle, a crucial regulator of metabolism. The supporting evidence presented is **compelling**, and the work is anticipated to captivate a wide audience within the metabolism discipline due to its extensive appeal and by providing inspiration for further hypothesis-driven research.

## Introduction

Protein post-translational modifications such as phosphorylation enable cells to rapidly respond to environmental changes by modifying protein function at low metabolic cost (*Humphrey et al., 2015b*). As a result of this high metabolic efficiency, phosphorylation is involved in nearly all biological processes and is dysregulated in numerous complex diseases (*Needham et al., 2019b*). Advances in mass spectrometry-based phosphoproteomics – the unbiased identification and quantification of protein phosphorylation – have led to the discovery of more than 100,000 phosphosites, revealing

**eLife digest** When we eat, the pancreas releases a hormone called insulin, which helps our tissues absorb glucose. Insulin works by triggering a cascade of events in cells, which include adding chemical tags called phosphate groups at thousands of specific locations on proteins. This tag causes the changes needed to move glucose from the blood into cells and also regulates many other essential functions in the cell.

If this process stops working and the body becomes resistant to the effects of insulin, it can lead to type 2 diabetes. This can result from a complex combination of genetic and lifestyle factors, which are difficult to study systematically in people. An alternative approach to understand these influences is to study mice, which are commonly used to investigate metabolic diseases and have contributed to our understanding of the mechanisms of type 2 diabetes. Using carefully bred mice allows precise control of their genetics and environment, revealing the independent and joint effects of these factors.

Monitoring differences in the phosphate groups on proteins, van Gerwen et al. studied five distinct inbred mouse strains fed either an ordinary diet or one that was high in fat and sugar. Nearly half of the biochemical events triggered by insulin were altered by genetics on the ordinary diet. High-fat, high-sugar feeding also reshaped the pattern of phosphate tags depending on the mouse strain. By examining these cellular responses, van Gerwen et al. identified proteins that may regulate the insulin response in muscle cells. Increasing the activity of one of these enzymes reversed insulin resistance in skeletal muscle cells grown in the laboratory.

This research underscores the importance of genetics in controlling insulin responses and shaping the impact of environmental challenges. It establishes a new opportunity in personalised medicine, which seeks to understand how an individual's genetics combine with their lifestyle to shape health. Furthermore, it identifies potential new targets for treating insulin resistance, paving the way for future research to develop more effective diabetes treatments.

that the phosphoproteome comprises vast, interconnected phosphorylation networks (*Olsen et al., 2006*; *Bodenmiller et al., 2010*; *Needham et al., 2019a*; *Leutert et al., 2023*), rather than the textbook view of isolated, linear kinase cascades.

The insulin signalling network is among the most studied phosphorylation networks. Post-prandial increases in blood glucose stimulate pancreatic insulin secretion, coordinating a metabolic switch in target tissues like skeletal muscle and adipose (*Haeusler et al., 2018*). Insulin increases glucose uptake into these tissues by promoting translocation of the glucose transporter GLUT4 to the plasma membrane, and serves other functions like enhancing protein synthesis, downregulating lipid catabolism, and altering gene transcription (*Haeusler et al., 2018*). To coordinate these functions, insulin triggers a phosphorylation cascade primarily involving activation of the Ser/Thr kinase Akt, regulation of downstream kinases including mTORC1 and GSK3, and modulation of parallel signalling arms (*Haeusler et al., 2018*; *Gehart et al., 2010*; *Sylow et al., 2013*). Seminal phosphoproteomics studies demonstrated that this cascade regulates over a thousand phosphosites, with many still uncharacterised in insulin action (*Humphrey et al., 2013*; *Humphrey et al., 2015a*; *Krüger et al., 2008*). Insulin resistance – the failure of insulin to promote glucose uptake in its target tissues – is triggered by genetic and environmental factors such as family history of metabolic disease and high-calorie diets (*James et al., 2021*). Although insulin resistance is a major precursor of metabolic disease including type 2 diabetes, its mechanistic basis remains unresolved (*James et al., 2021*; *Fazakerley et al., 2019*; *van Gerwen et al., 2023*).

Interactions between genetics and environment significantly regulate biomolecular processes, including insulin resistance (*Civelek and Lusis, 2014*; *Hunter, 2005*; *Nelson et al., 2022*; *Montgomery et al., 2013*). As signalling pathways connect the extracellular environment to intracellular proteins, they are likely a major conduit of gene-by-environment interactions. Yet, how global phosphorylation signalling networks are regulated across different genetic backgrounds is relatively unexplored. Recent phosphoproteomics studies in yeast (*Grossbach et al., 2022*) and mice (*Zhang et al., 2023*) identified genetic variants affecting multiple phosphosites, but did not analyse the phosphoproteome's response to acute perturbation, which is crucial to its role as a signal transduction system. We have also shown marked variation in acute signalling responses to exercise or insulin across individuals

(*Needham et al., 2022*). However, this study did not systematically assess the relative contributions of genetics and the environment (*Needham et al., 2022*). Understanding how these variables intersect with signal transduction is fundamental to our basic knowledge of signalling and the advancement of personalised medicine, which advocates individualised treatment regimens based on genetic risk factors and gene-by-environment interactions (*Hunter, 2005*; *McCarthy et al., 2008*).

Inbred mice allow precise control of genetics and environment unachievable in human studies, enabling examination of how these factors interact to influence biomolecular systems (*Lusis et al., 2016*). Here, we performed phosphoproteomics on insulin-stimulated or control skeletal muscle from five genetically distinct inbred mouse strains fed either an ordinary chow diet or a high-fat, high-sugar 'western style' diet. Strikingly, we found that genetic background influenced both the phosphoproteomic insulin response of chow-fed mice, as well as how these responses were modified by high-fat, high-sugar feeding. These signalling changes were reflected in altered activity profiles of multiple kinases and provided insight into the functional organisation of the insulin signalling network by revealing subnetworks of coregulated phosphosites. A major challenge in phosphoproteomics studies is pinpointing important regulatory events among the many responding to a stimulus. We reasoned that associating changes in protein phosphorylation across the gene-by-environment landscape with phenotypic change – in this case insulin-stimulated glucose uptake – would dissect mechanistic targets with greater fidelity. This approach generated known as well as candidate regulators of insulin-stimulated glucose uptake, leading us to demonstrate that glycolytic upregulation reverses insulin resistance. Our work represents the first global portrait of insulin signalling network plasticity in response to genetic and environmental variation, which will serve as an important resource in future studies of insulin action and resistance.

## Results

### Phosphoproteomics of insulin signalling in mouse skeletal muscle

To study how protein phosphorylation networks are affected by genetics and environment, we examined insulin signalling in five genetically distinct inbred mouse strains including four lab strains with diverse metabolic phenotypes (C57Bl6J, NOD, BXH9, and BXD34) (*Nelson et al., 2022*), and the wild-derived CAST strain (*Figure 1a*). Mice underwent a 6-week diet regimen of standard lab diet (CHOW) or a high-fat high-sucrose diet (HFD), which is commonly used to induce insulin resistance (*Nelson et al., 2022*; *Burchfield et al., 2018*). Consistent with their diverse genetics, these strains differed in morphometric parameters (body weight, adiposity, lean mass) and metabolic traits (fasting blood glucose, fasting blood insulin, glucose tolerance) both on the CHOW diet and in their response to HFD-feeding (*Figure 1—figure supplement 1*).

We focused on skeletal muscle, as it is the site of greatest post-prandial glucose uptake and the most significant contributor to impaired glucose disposal in type 2 diabetes (*DeFronzo, 1988*). Specifically, we chose to examine the soleus muscle, because its largely oxidative fibre composition resembles human muscle tissue more than other murine muscles (*Schiaffino and Reggiani, 2011*). Mice were injected retro-orbitally with saline control or insulin for 10 min, and the soleus was collected for phosphoproteomic analysis (*Figure 1a*). A tritiated 2-deoxyglucose ($^3$H-2DG) tracer was co-injected to measure soleus glucose uptake.

Using the EasyPhos workflow and data-independent acquisition (DIA) mass spectrometry (*Humphrey et al., 2018*; *Bekker-Jensen et al., 2020*), we quantified 28,809 phosphopeptides across 95 biological samples, corresponding to 23,126 unique high-confidence phosphosites (class I; localisation score >0.75) on 3507 proteins (*Figure 1b*, *Supplementary file 1*). On average, we quantified 15,395 phosphopeptides in each sample (*Figure 1—figure supplement 2a*). Due to the range in soleus mass across strains (*Figure 1—figure supplement 1d*) we altered the protein material used for EasyPhos (C57Bl6J and NOD: 755 μg, BXH9 and BXD34: 511 μg, CAST: 364 μg), though phosphopeptide quantification was minimally affected, with only 12.4% fewer phosphopeptides quantified on average in CAST compared to the C57l6J/NOD (average 13,891.56 vs 15,851.29 *Figure 1—figure supplement 2a*). Furthermore, while different strains clustered noticeably by the amount of protein material used, samples from animals of the same strain and diet were still highly correlated and generally clustered together, implying the data are reproducible (*Figure 1—figure supplement 2b–d*).

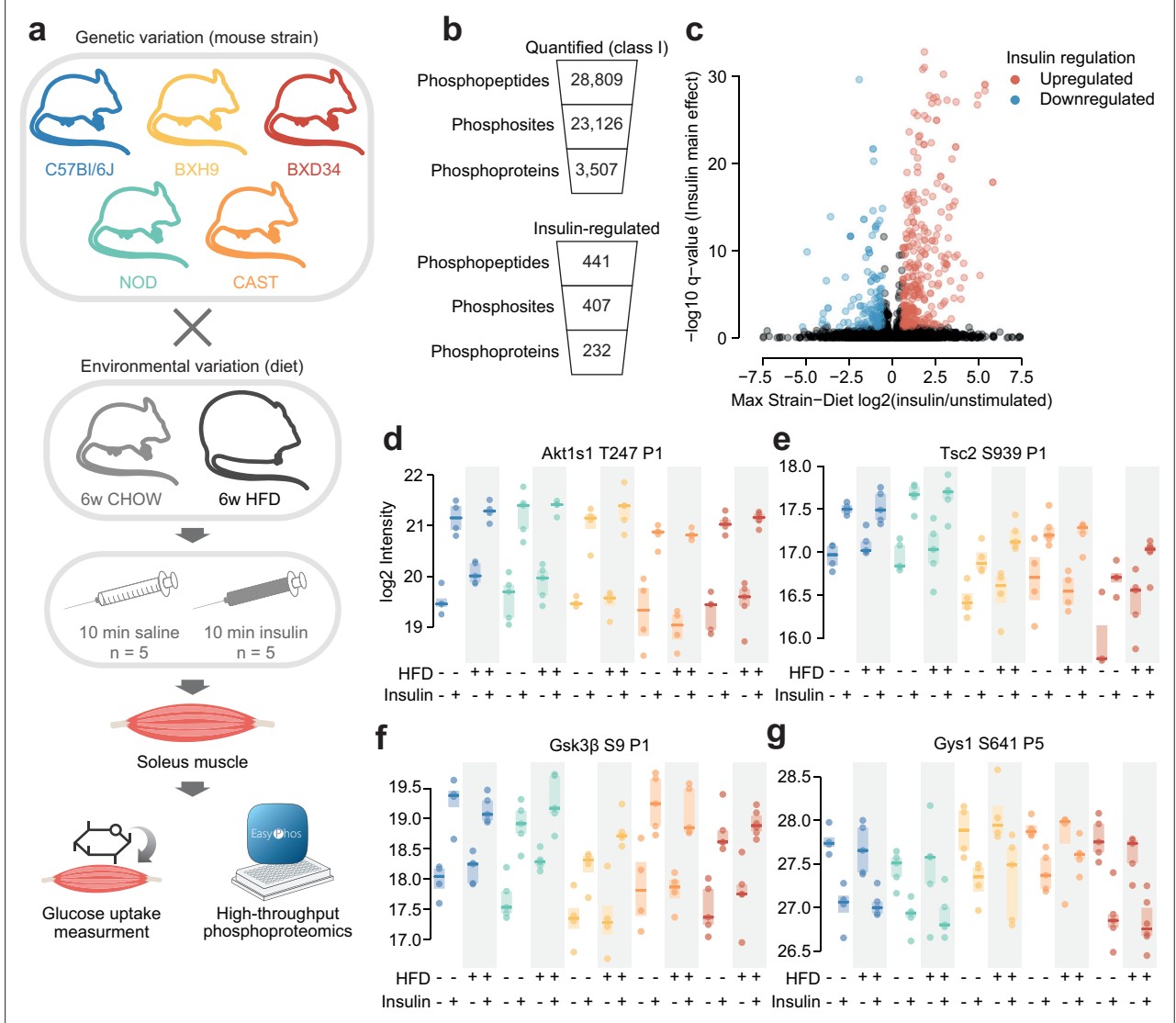

**Figure 1.** Phosphoproteomics of insulin signalling in mouse skeletal muscle. (**a**) Workflow for skeletal muscle phosphoproteomics of insulin signalling. (**b**) Quantification of skeletal muscle phosphoproteomics. (**c**) Volcano plot identifying insulin-regulated phosphopeptides. The greatest log2(insulin/unstimulated) fold change across strain-diet combinations is plotted against significance (insulin stimulation main effect, three-way ANOVA). Three phosphopeptides with -log10 q-values greater than 35 were removed for visual clarity. (**d–g**) Example insulin-regulated phosphopeptides. The protein and phosphorylated amino acid are indicated, as well as the number of phosphosites on the phosphopeptide (e.g. 'P1'). $n$=4–6 biological replicates.

The online version of this article includes the following figure supplement(s) for figure 1:

**Figure supplement 1.** Genetics and diet alter morphometric and metabolic phenotypes.

**Figure supplement 2.** Quality control analysis of phosphoproteomics data.

**Figure supplement 3.** Characterisation of the insulin-regulated phosphoproteome.

To explore the soleus insulin signalling network, we examined phosphopeptides altered by insulin stimulation in at least one of the ten strain-diet combinations. First, to allow comparison across conditions, phosphopeptides were retained if they were quantified well enough to assess the effect of insulin in more than eight strain-diet combinations (≥3 insulin-stimulated values and ≥3 unstimulated values in each combination). We then tested the resulting 10,432 highly quantified phosphopeptides for significant differences between unstimulated and insulin-stimulated samples (three-way ANOVA insulin main effect q-value <0.05) that were of sufficient magnitude in at least one strain-diet combination (insulin/unstimulated fold change >1.5). This resulted in 441 insulin-regulated phosphopeptides on 232 proteins, which is noticeably more than recent studies of 10 min insulin signalling in

patient-derived myoblasts (174 phosphopeptides; *Batista et al., 2020*) and 242 phosphopeptides (*Haider et al., 2021*) or mouse adipose tissue (319 phosphopeptides; *Fazakerley et al., 2023*; *Figure 1b and c*; *Supplementary file 1*).

Our analysis recovered many well-studied insulin-regulated phosphosites, including Akt substrates such as T247 on Akt1s1 (PRAS40), S939 on Tsc2, and S9 on Gsk3β (*Figure 1d–f*), as well as targets of downstream kinases including the Gsk3β substrate S641 on Gys1 (*Figure 1g*). Enrichment of Gene Ontology (GO) biological processes recapitulated canonical insulin signalling axes including 'insulin receptor signalling pathway', 'phosphatidylinositol 3-kinase signalling', 'protein kinase B signalling', and 'TORC1 signalling', and multiple pathways related to glucose metabolism, fatty acid metabolism, autophagy, and protein translation, reflecting known targets of insulin action (*Figure 1—figure supplement 3a*). Furthermore, insulin-regulated phosphosites were enriched for insulin regulation in our previous human skeletal muscle phosphoproteome (fold enrichment = 4.22, p=$9.50 \times 10^{-24}$, one-sided Fisher's exact test, *Figure 1—figure supplement 3b*; *Needham et al., 2022*). Despite this, only half of all insulin-regulated phosphopeptides (228/441) were previously annotated as insulin-regulated in the PhosphositePlus database (*Hornbeck et al., 2015*; *Figure 1—figure supplement 3c*), highlighting the potential of our data to discover novel aspects of insulin signalling while recapitulating known components. In addition to insulin, exercise also promotes GLUT4 translocation in skeletal muscle (*Klip et al., 2019*). We identified a small subset of phosphosites regulated by insulin in this study that were also regulated by exercise in two separate human phosphoproteomics studies (*Needham et al., 2022*; *Hoffman et al., 2015*; *Figure 1—figure supplement 3d*; *Supplementary file 2*, phosphosites: Eef2 T57 and T59, Mff S129 and S131, Larp1 S498, Tbc1d4 S324, Svil S300, Gys1 S645), providing a starting point for exploring conserved signalling regulators of GLUT4 translocation. Overall, our phosphoproteomics data provide a comprehensive and high-quality atlas of insulin signalling in mouse skeletal muscle.

## Genetics and diet modulate insulin signalling

The influence of genetic and environmental variation on global insulin signalling responses is largely unknown. We therefore developed a pipeline to address this question using our phosphoproteomics data (*Figure 2a*). First, we converted the intensity values of each insulin-regulated phosphopeptide to 'insulin response' values, by normalising insulin-stimulated data to the unstimulated median of the corresponding strain-diet combination. Since protein expression should not change within a 10 min insulin stimulation (*Fazakerley et al., 2023*), this allowed us to parse out protein abundance differences across strains and diets and focus solely on acute signalling processes. We then assessed the impact of genetics in CHOW-fed mice by identifying phosphopeptides with differing insulin responses in one or more strains compared to C57Bl6J ('Strain effect'). Lastly, we explored the effects of HFD-feeding on signalling as two types of 'Diet effects', either as a 'Uniform diet effect' – where HFD-feeding affects each strain similarly – or a 'Strain×Diet effect' – where its impact depends on the mouse strain. Analyses of 'Strain effects' and 'Diet effects' were performed separately, so a phosphopeptide could have both a Strain effect and a Strain×Diet/Uniform diet effect.

Almost half of the 441 insulin-regulated phosphopeptides displayed a Strain effect (*Figure 2b*, *Supplementary file 1*). These included phosphopeptides where C57Bl6J had a stronger insulin response than other strains, such as S15 on the RNA methyltransferase Rnmt (*Figure 2c*), and phosphopeptides where C57Bl6J had a weaker insulin response, such as S48 on the vesicle fusion regulator Vamp3 (*Figure 2d*). Vamp3 S48 is predicted to be highly functional (functional score = 0.750) (*Ochoa et al., 2020*), its phosphorylation correlates with glucose uptake in insulin-stimulated and/or exercised human skeletal muscle (*Needham et al., 2022*), and Vamp3 overexpression rescues GLUT4 translocation in insulin resistance (*Schwenk et al., 2012*), suggesting that this site may represent a genetically variable control point of GLUT4 trafficking. In general, insulin responses were weaker in the four remaining strains compared to C57Bl6J, though the extent of this trend was strain-dependent (*Figure 2e*). In all, the strain-affected phosphopeptides reveal a unique fingerprint of insulin signalling within each strain (*Figure 2f*), highlighting the complex and widespread effects of genetic variation on signalling networks.

We next examined the impact of HFD-feeding in insulin signalling. Strain×Diet effects were more prevalent than Uniform diet effects (110 vs 10 phosphopeptides, *Figure 2g*; *Supplementary file 1*), suggesting that the molecular impact of dietary perturbation was strongly modulated by genetic

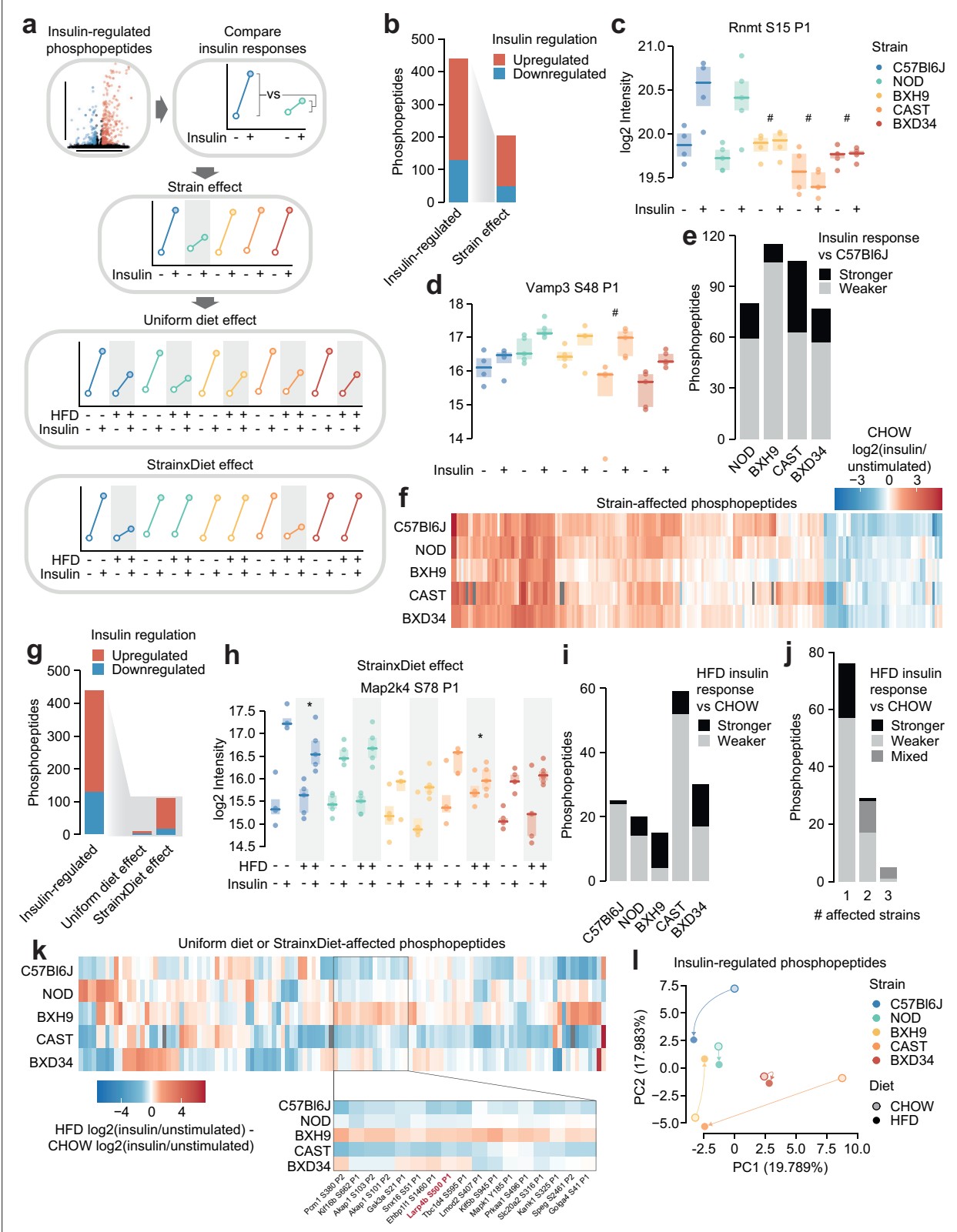

**Figure 2.** Genetics and diet modulate insulin signalling. (**a**) Schematic for identifying effects of strain and diet on insulin signalling. (**b**) The number of total insulin-regulated phosphopeptides and those with a Strain effect. (**c–d**) Two phosphopeptides with Strain effects. ANOVAs were performed on CHOW insulin responses following two-sided t-tests comparing each strain to C57Bl6J (q-values: #). Only CHOW values are shown. (**e**) The number of phosphopeptides with stronger or weaker insulin regulation in each strain compared to C57Bl6J. (**f**) Heatmap displaying all insulin-regulated

*Figure 2 continued on next page*

Figure 2 continued

phosphopeptides with a Strain effect. Missing values are coloured grey. (**g**) The number of total insulin-regulated phosphopeptides and those with diet effects. (**h**) A phosphopeptide with a Strain×Diet effect. A two-way ANOVA was performed on insulin response values followed by two-sided t-tests comparing HFD to CHOW within each strain (q-values: *). (**i–j**) The number of phosphopeptides with a Strain×Diet effect in (**i**) each strain, or (**j**) each number of strains. Colour indicates whether the insulin response in HFD is stronger vs CHOW, weaker vs CHOW, or both in different strains ('Mixed'). (**k**) Heatmap displaying all insulin-regulated phosphopeptides with a Uniform diet effect or Strain×Diet effect. Inset displays example sites where BXH9 effects contrasted other strains. (**l**) Principal component analysis (PCA) of all insulin-regulated phosphopeptides using the log2(insulin/unstimulated) fold changes for each strain-diet combination. The percentage of total variance explained by each principal component is indicated. */#: 0.01≤q<0.05, **/##: 0.001≤q<0.01, ***/###: q<0.001. $n$=4–6 biological replicates.

background. Strain×Diet effects impacted known regulatory phosphosites such as the inhibitory site S78 on Map2k4, whose insulin response was attenuated by HFD-feeding only in C57Bl6J and CAST (*Figure 2h*). Map2k4 activates p38 and JNK kinases which have been implicated as drivers of HFD-induced insulin resistance (*Gehart et al., 2010*), and based on the behaviour of S78, the orchestration of this detrimental signalling axis in HFD-feeding may depend on genetic background.

HFD-feeding exerted complex effects on signalling, with the balance between suppressed and enhanced insulin responses varying across strains (*Figure 2i*). Furthermore, insulin-regulated phosphopeptides were largely altered by HFD-feeding in only a single strain, and when multiple strains were affected, they often changed in opposite directions (*Figure 2j–k*). For instance, multiple insulin responses were strengthened in BXH9 but weakened in C57Bl6J or CAST (*Figure 2k*), such as S500 on the translation regulator Larp4b (*Figure 1—figure supplement 3e*). Principal component analysis supported the highly divergent impact of dietary perturbation, as HFD-feeding displaced each strain in a distinct direction in principal component space (*Figure 2l*). Strain×Diet and Strain effects were driven by a mixture of changes to insulin-stimulated phosphorylation, unstimulated phosphorylation, or both, highlighting the complexity of these signalling alterations (*Figure 1—figure supplement 3f–g*). This analysis demonstrates the pervasive role of genetics in shaping signalling networks, as genetic background profoundly modulated the effect of HFD-feeding on insulin signalling.

## Exploring genetic and dietary modulation of the insulin signalling network

To understand the insulin signalling circuitry and functional pathways modulated by genetics and diet, we curated a network of 160 insulin-regulated phosphosites, comprising sites from a knowledge pathway-derived list of canonical insulin signalling proteins (*Needham et al., 2022*) and substrates of insulin-regulated kinases (*Figure 3*, see Methods). Strain and diet affected multiple highly studied signalling sites, including Tsc2 S939 (Diet effect), Gsk3α S21 (Strain and Diet effects), and Tbc1d4 T649 (Strain and Diet effects), while other sites such as Gsk3β S9 and Akt1s1 T247 were unaffected. Interestingly, strain and diet affected both canonical and non-canonical insulin signalling proteins to a similar extent (*Figure 1—figure supplement 3h–i*). Non-canonical phosphosites could shed light on underappreciated outcomes of insulin action either altered or unaffected by genetics and the environment, such as the p70S6K substrate S47 on Dnajc2 (Strain and Diet effects), which drives cellular senescence (*Barilari et al., 2017*), and S315 on Pcyt1a (no Strain or Diet effect), which inhibits phosphatidylcholine biosynthesis (*Agassandian et al., 2005*; *Figure 3*).

No functional pathways were overrepresented within strain or diet-affected phosphosites relative to all insulin-regulated sites, implying that genetics and environment modulate most or all outcomes of insulin. For instance, strain and diet affected regulatory phosphosites controlling vesicle trafficking (S348, T575, S595, and T649 on the GLUT4 trafficking regulator Tbc1d4); glucose metabolism (S469 and S486 on Pfkfb2, which promote glycolysis); mitochondrial structure and function (S129 on Mff, which promotes mitochondrial fission); autophagy (S555 on the master autophagy regulator Ulk1); gene transcription (the inhibitory site S108 on the transcription factor Tfeb); and mRNA translation (S236 on ribosomal proteins S6, S422 on Eif4b, and S64 and T69 on Eif4ebp1, which promote translation). Interestingly, Strain and Diet effects overlapped significantly (fold enrichment = 1.50, p=4.40 × 10⁻⁹, two-sided Fisher's exact test, *Figure 1—figure supplement 3j*), implying some phosphosites may be more amenable to regulation overall. As a notable exception, all six insulin-regulated phosphosites on Plin1 had Diet effects while only one had a Strain effect (*Figure 3*). Plin1 coats and regulates lipid droplets, hence this enrichment of Diet effects may represent a signalling response

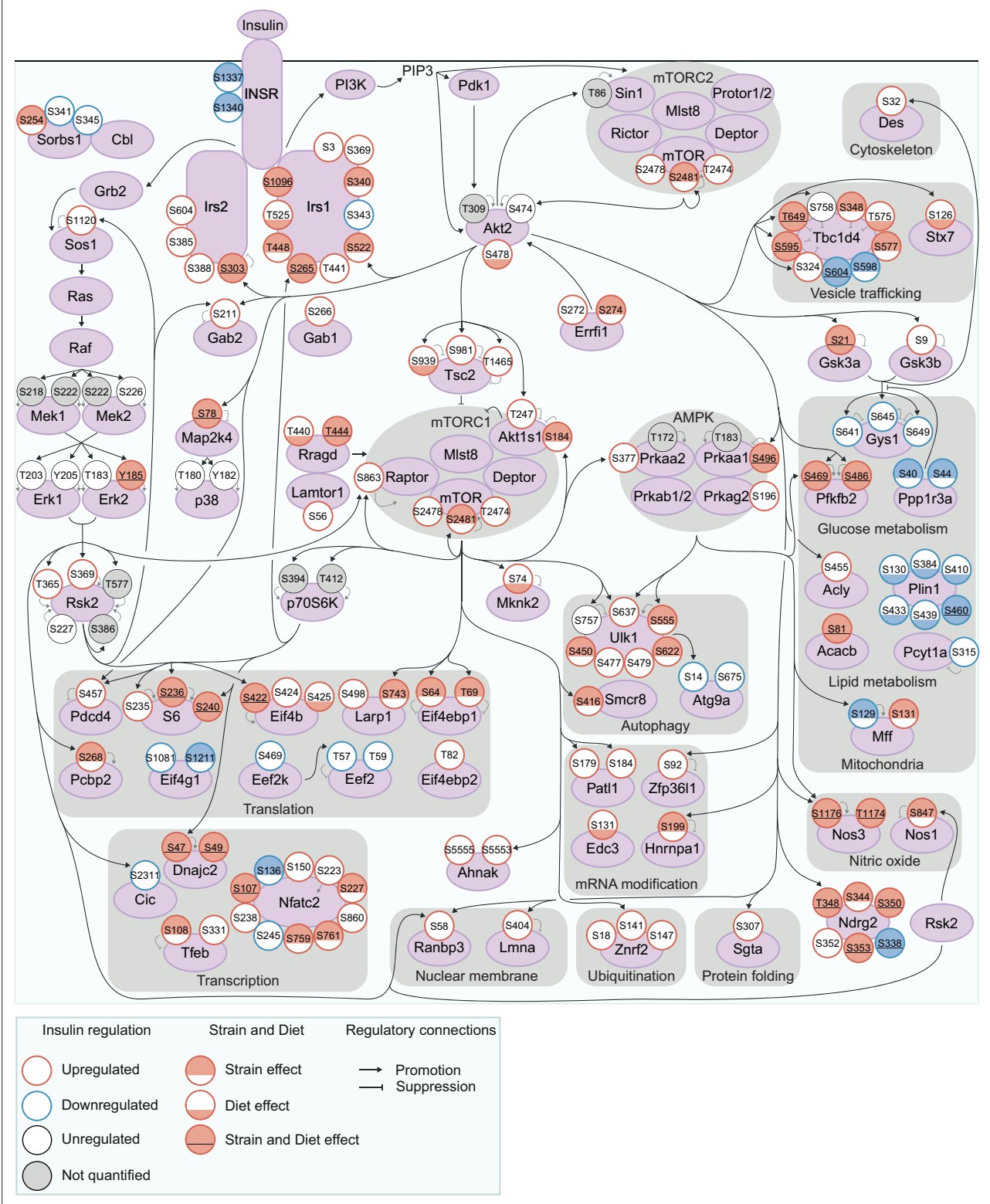

**Figure 3.** Exploring genetic and dietary modulation of the insulin signalling network. A curated network of 160 insulin-regulated phosphosites. Phosphosites are depicted as circles where the outline colour denotes the direction of insulin regulation, and the inner colour denotes the presence of Strain effects or Diet effects (either a Strain×Diet or Uniform diet effect). Black arrows indicate regulatory relationships from proteins to other proteins or phosphosites. Grey lines indicate phosphosite regulatory roles.

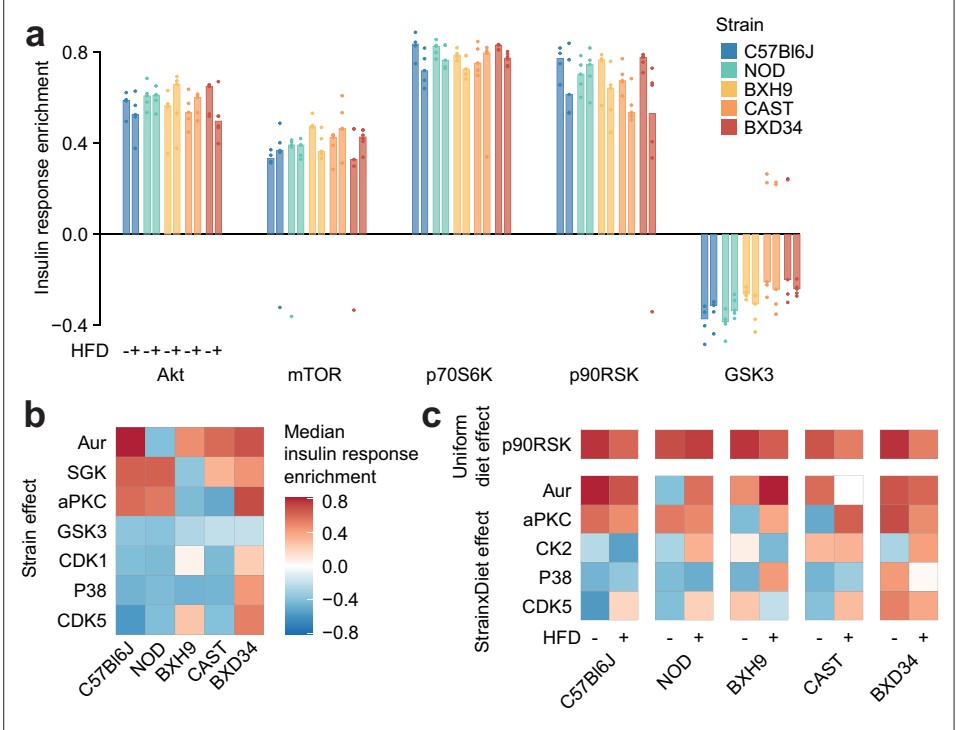

**Figure 4.** Genetics and diet rewire insulin-regulated kinase signalling. (**a**) Kinase substrate enrichment analysis (KSEA) (*Hernandez-Armenta et al., 2017*) of five canonical insulin-regulated kinases using insulin response values and kinase substrate annotations from PhosphositePlus (*Hornbeck et al., 2015*). (**b–c**) Kinase enrichment scores were tested for (**b**) Strain effects (CHOW ANOVA adjusted p<0.05) or (**c**) Strain×Diet effects (two-way ANOVA interaction effect adjusted p<0.05) and Uniform diet effects (Diet main effect adjusted p<0.05, interaction effect adjusted p≥0.05). n=4–6 biological replicates.

to increased intramuscular lipids in the HFD condition. Overall, genetics and environment triggered widespread alterations in insulin signalling impinging on diverse cellular pathways.

## Genetics and diet rewire insulin-regulated kinase signalling

The extensive signalling changes caused by genetics and diet may result from altered kinase regulation. We tested this hypothesis using a kinase substrate enrichment analysis (KSEA) (*Hernandez-Armenta et al., 2017*) on phosphopeptide insulin responses. KSEA accurately captured the activation of canonical insulin-regulated kinases (Akt, mTOR, p70S6K, and p90RSK) and the deactivation of GSK3, confirming the validity of the approach (*Figure 4a*). Focussing on CHOW-fed mice, we identified seven kinases differentially enriched across mouse strains (ANOVA adjusted p-value <0.05, *Figure 4b*). For example, insulin activated SGK and deactivated GSK3 more in C57Bl6J and NOD than in other strains (*Figure 4b*). Extending this analysis to all mice, we identified kinases with Uniform diet or Strain×Diet effects (*Figure 4c*). Akin to our analysis of individual phosphosites (*Figure 2*), Strain×Diet effects were more prevalent than Uniform diet effects (five kinases compared to one), indicating that genetic background strongly influences the impact of HFD-feeding on kinase signalling. These results suggest that the observed phosphosite signalling changes could be partly due to altered insulin regulation of multiple kinases.

## Biological variation reveals functional organisation of the insulin signalling network

KSEA predicted changes in overall kinase activity, but we questioned if substrates of the same kinase could be differentially regulated by genetic and environmental variation. As a case study we examined substrates of Akt – a master regulator of insulin signal transduction – to assess the similarity of their insulin responses across strains and diets. Strikingly, we observed a range of both positive

and negative correlations (*Figure 5a*). For instance, while Tsc2 S939 and Akt1s1 T247 both activate mTORC1, their insulin responses correlated poorly (*r*=0.202, p=0.168, *Figure 5b*). Supporting these findings, similar heterogeneity in Akt substrate insulin responses has previously been observed in skeletal muscle from humans with differing metabolic health (*Tonks et al., 2013*). Hierarchical clustering revealed four distinct groups of positively correlated Akt substrates (*Figure 5a*), suggesting these groups may coordinate distinct functional outputs of Akt signalling.

We next explored if the genetic and environmental variation in our study could reveal organisational principles of the entire insulin signalling pathway. By performing weighted gene correlation network analysis (WGCNA; *Zhang and Horvath, 2005*; *Langfelder and Horvath, 2008*), we identified eight subnetworks of coregulated insulin-responsive phosphopeptides (*Figure 5c*) varying in size from 16 to 120 phosphopeptides, with 91 assigned to no subnetwork (*Figure 5d*, *Supplementary file 3*). Examining the subnetwork 'eigenpeptides' – a weighted average of the constituent phosphopeptides (*Zhang and Horvath, 2005*; *Langfelder and Horvath, 2008* )– revealed that the subnetworks captured distinct effects of genetics and diet on insulin signalling (*Figure 5e*; *Figure 5—figure supplement 1a*). For example, HFD-feeding attenuated the insulin response of subnetwork I in CAST and C57Bl6J strains (t-test adjusted p=0.0256, 0.0365), while subnetwork II was affected by HFD-feeding only in CAST and NOD (*Figure 5e*; *Figure 5—figure supplement 1a*, t-test adjusted p=0.00258, 0.0256). This suggests that the subnetworks may be sensitive to distinct cellular information.

Next, we characterised the regulatory and functional nature of these subnetworks. Canonical insulin-regulated kinases such as Akt and mTOR were enriched across multiple subnetworks (*Figure 5f*), confirming that genetic and environmental variation can reveal uncoupling of substrates targeted by the same kinase (*Figure 5a–b*). Nevertheless, visualising these subnetworks within our curated insulin signalling pathway (*Figure 3*) revealed cases where signal flowed through a single subnetwork, such as from Erk2 (Y185) to its target kinase Rsk2 (T365 and S369) and Rsk2 substrates (Gab2 S211 and Nos1 S847) within subnetwork III (*Figure 5—figure supplement 2*). Within multipl phosphorylated proteins, phosphosites either belonged to the same subnetwork (e.g. Plin1) or diverse subnetworks (e.g. Tbc1d4 and the transcription factor Nfatc2), suggesting the latter may serve as hubs integrating diverse cellular information (*Figure 5—figure supplement 2*).

Some subnetworks were enriched in specific cellular compartments (GO), implying that common localisation may facilitate coregulation of phosphosites (*Figure 5g*). Coregulation may partition functional outcomes of insulin action, as certain biological processes were enriched only in select subnetworks (*Figure 5h*). These included known insulin targets like 'negative regulation of lipid catabolic process' in subnetwork I and 'positive regulation of glycogen biosynthetic process' in I, VI, and VIII (*Figure 5h*). To further probe functional differences, we analysed phosphopeptide subnetwork membership scores, which revealed additional pathways enriched in individual subnetworks. However, these results were not significant after p-value adjustment and hence are suggestive only (*Figure 5—figure supplement 1b–d*). Lastly, we leveraged our previous phosphoproteomic time course of insulin signalling to interrogate subnetwork dynamics (*Humphrey et al., 2013*) and found that phosphopeptide insulin response dynamics varied across subnetworks (*Figure 5i*). This reveals distinct temporal regulation as another feature underlying the substructure of the insulin signalling network. Overall, genetic and environmental diversity illuminated the complex coregulation structure of insulin signalling, featuring subnetworks that evade known network circuitry and present unique functional signatures.

## Leveraging biological variation to identify drivers of insulin responsiveness

We have so far described the marked influence of genetic background and HFD-feeding on skeletal muscle insulin signalling, evident at the level of individual phosphosites, protein kinases, and coregulated network modules. We hypothesised that by associating this signalling diversity with a phenotypic output of insulin, such as enhanced glucose uptake, we would filter out mechanistically irrelevant phosphosites and hence prioritise molecular regulators of the phenotype. To test this hypothesis, we measured in vivo glucose uptake with [3]H-2DG tracer in the same muscle samples used for phosphoproteomics. Insulin-stimulated glucose uptake differed by more than twofold across strains (two-way ANOVA strain effect p=4.78 × 10[−7]) and was almost uniformly decreased by HFD-feeding (two-way

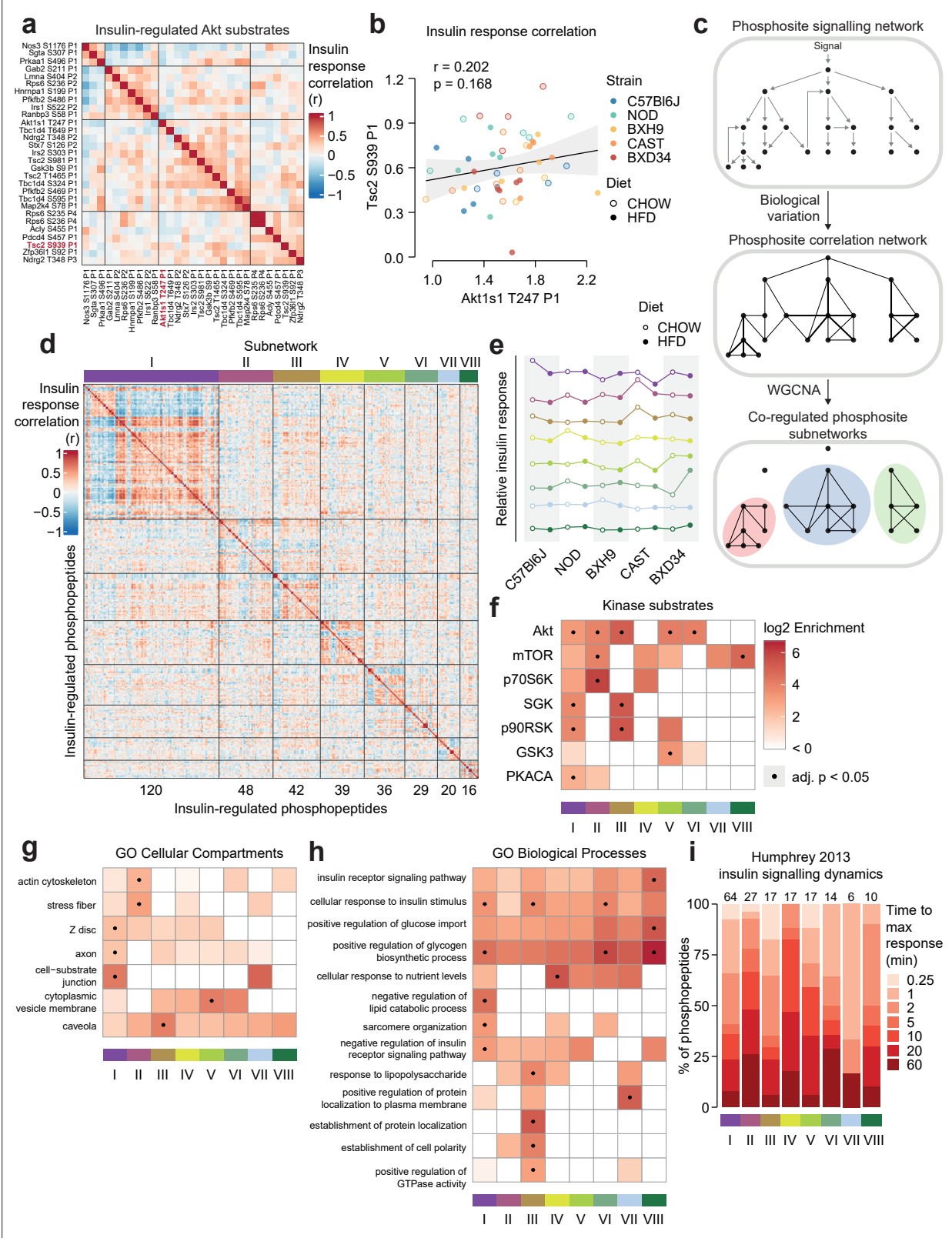

**Figure 5.** Biological variation reveals functional organisation of the insulin signalling network. (**a**) Pairwise Pearson's correlation of the insulin response values of insulin-regulated Akt substrates. Substrates were separated into four clusters by hierarchical clustering followed by tree cutting. (**b**) The correlation between insulin response values of the Akt substrates Tsc2 S939 and Akt1s1 T247. Linear regression is indicated with 95% confidence intervals. (**c**) Rationale for performing weighted gene correlation network analysis (WGCNA). (**d**) Pairwise Pearson's correlation of all insulin-regulated

*Figure 5 continued on next page*

*Figure 5 continued*

phosphopeptides separated into WGCNA-derived subnetworks. The number of phosphopeptides in each subnetwork is indicated below the heatmap. (**e**) The 'eigenpeptide' of each subnetwork. The median of each strain-diet combination is shown. (**f–h**) The enrichment of (**f**) PhosphositePlus-derived kinase substrate annotations (*Hornbeck et al., 2015*), (**g**) Gene Ontology (GO) cellular compartments, and (**h**) GO biological processes within each subnetwork relative to the entire phosphoproteome (one-sided Fisher's exact test, Benjamini-Hochberg p-value adjustment). (**i**) The time taken for phosphopeptides to reach maximum insulin-stimulated intensity in a previous study of insulin signalling dynamics (*Humphrey et al., 2013*). The number of phosphopeptides mapped into the study is indicated above each bar.

The online version of this article includes the following figure supplement(s) for figure 5:

**Figure supplement 1.** Characterising insulin signalling subnetworks.

**Figure supplement 2.** Organisation of insulin signalling subnetworks.

ANOVA diet effect $p = 1.83 \times 10^{-5}$) (*Figure 6a*). This highlights that genetic background and dietary status are key determinants of insulin responsiveness.

To prioritise signalling nodes responsible for differences in insulin responsiveness, we correlated all insulin-regulated phosphopeptides with glucose uptake in insulin-stimulated muscles, resulting in 37 significantly correlated phosphopeptides ($r > 0.35$ or $< -0.35$, q-value $<0.1$, *Figure 6b*). The most significantly correlated phosphopeptide contained T1174 and S1176 on the nitric oxide synthase Nos3. The latter serves as a positive control for our analysis, as this site is known to be phosphorylated in response to insulin to promote in vivo glucose uptake by vasodilation (*Roy et al., 1998*; *Bahadoran et al., 2020*; *Dimmeler et al., 1999*; *Figure 6c*). Other correlated phosphopeptides that could modulate insulin responsiveness include S1751 on Afdn, a phosphosite implicated in insulin action (*Lundh et al., 2019*), and S196 on the Prkag2 subunit of AMPK, a major metabolic signalling hub promoting glucose uptake (*O'Neill, 2013 Figure 6b*). These examples suggest that our analysis prioritised regulators of glucose uptake.

While the above analysis identified phosphosites associated with glucose uptake through their absolute abundance, we hypothesised that for some phosphosites, the magnitude of their response to insulin may be a stronger determinant of insulin action. We found that the insulin response values of 13 phosphopeptides correlated with insulin-stimulated glucose uptake ($r > 0.35$ or $< -0.35$, q-value $<0.1$, *Figure 6d*). These were largely distinct from the 37 phosphopeptides identified in our first analysis, indicating that the two approaches captured complementary information. Several of these phosphopeptides could regulate insulin-stimulated glucose uptake, such as the regulatory site S469 on the enzyme Pfkfb2 which activates glycolysis, a major pathway for glucose consumption (*Deprez et al., 1997*; *Marsin et al., 2000*; *Ros and Schulze, 2013*; *Figure 6e*), and S177 on Rcsd1, which could affect GLUT4 vesicle transport via actin cytoskeleton remodelling (*Stöckli et al., 2011*; *Figure 6d*). The associations identified in these analyses could arise uniquely at the level of protein phosphorylation or could be driven by changes in total protein abundance. To assess this, we mined soleus total proteomics data from a recent study of seven CHOW and HFD-fed mouse strains, three of which were in common with this study (C57BL6J, BXH9, BXD34) (*Nelson et al., 2022*). Of the 34 glucose uptake-associated phosphoproteins we identified, 16 were quantified in both studies and only two additionally correlate with glucose uptake at the total protein level (Ppp6r1 and Ttn, *Figure 6—figure supplement 1a*). Hence, it seems likely that most glucose uptake-phosphosite associations were not driven by protein abundance changes, underscoring the utility of phosphoproteomics to provide unique information on top of more classical omics layers.

In addition to individual phosphosites, the status of larger signalling network components could also influence insulin responsiveness. Kinase enrichment scores affected by strain or diet did not correlate with glucose uptake (*Supplementary file 4*), suggesting insulin action is not dominated by the net activity of specific kinases. Interestingly, two WGCNA-derived insulin signalling subnetworks correlated with glucose uptake: subnetwork V ($r = 0.436$, $p = 0.00173$) and subnetwork I ($r = 0.332$, $p = 0.0197$, *Figure 6f*). Subnetwork V could modulate glucose uptake through actin cytoskeleton remodelling via Rscd1 S177, through glucose metabolism promotion via Gys1 S641 (*Supplementary file 3*), and by influencing GLUT4 vesicle trafficking due to its enrichment at 'cytoplasmic vesicle membranes' (*Figure 5g*). It was also enriched in substrates of GSK3, which has been implicated in insulin resistance in skeletal muscle (*Dokken et al., 2005*; *Henriksen and Teachey, 2007*; *Ring et al., 2003*) and adipose tissue (*Fazakerley et al., 2023*). Subnetwork I, the largest cluster containing 27% of insulin-regulated phosphopeptides, was enriched in multiple kinases and biological processes

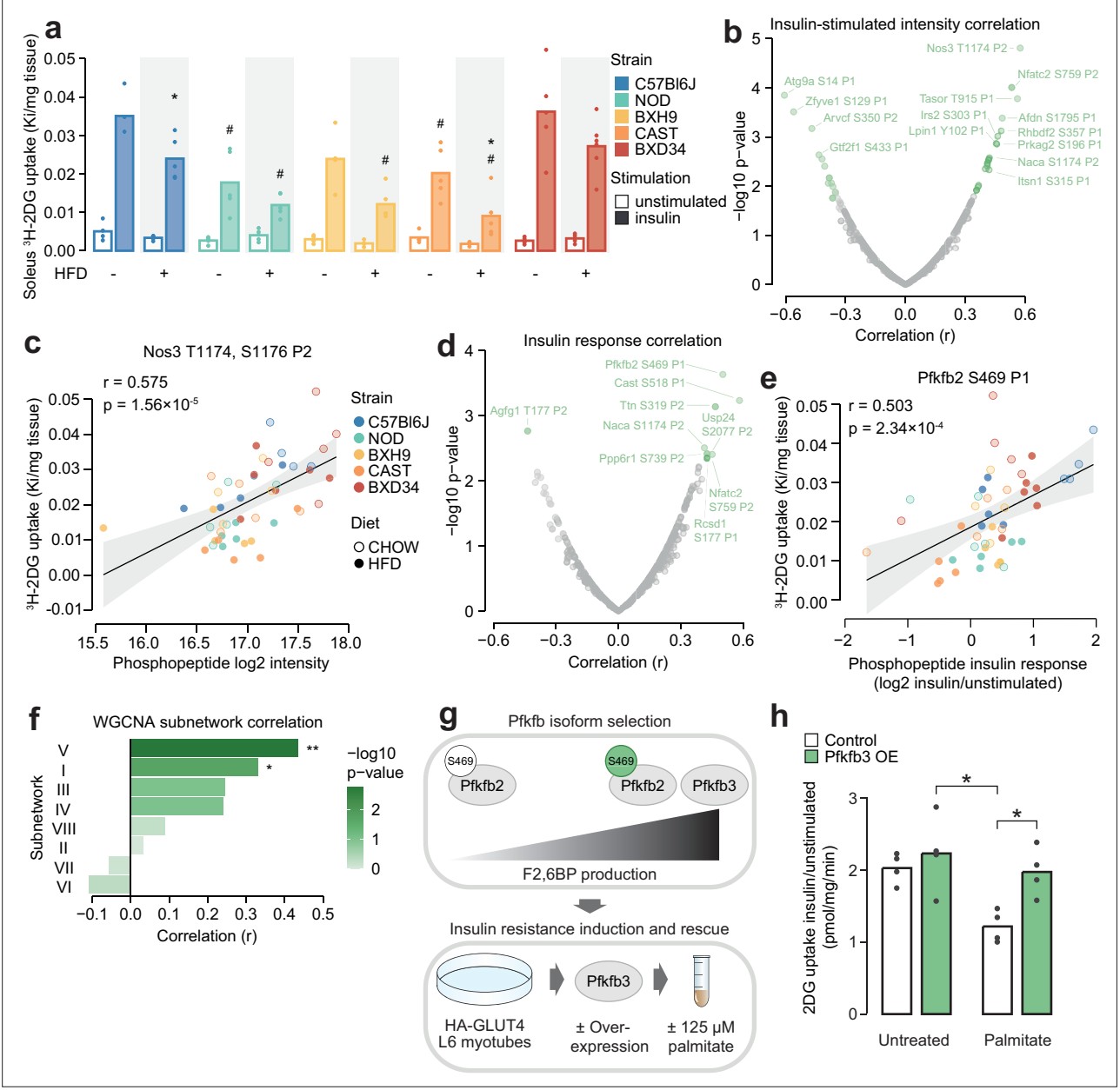

**Figure 6.** Leveraging biological variation to identify drivers of insulin responsiveness. (**a**) The uptake of ³H-2DG into mouse soleus muscle after a 10 min injection of insulin (1 U/kg lean mass; 'insulin') or saline ('unstimulated') calculated as a rate constant (Ki). Two-sided t-tests were performed on insulin-stimulated uptake values to compare HFD to CHOW within each strain (adjusted p-value: *) or each strain to C57Bl6J within either diet (adjusted p-value: #). n=4–6 biological replicates. (**b**) Pearson's correlation between log2 intensity of insulin-regulated phosphopeptides and ³H-2DG uptake within insulin-stimulated mice. Significantly correlated phosphopeptides (q-value <0.1, r>0.35 or r<–0.35) are coloured green and select correlated phosphopeptides are labelled. (**c**) Correlation of Nos3 T1174, S1176 insulin-stimulated intensity with insulin-stimulated ³H-2DG uptake. Linear regression is indicated with 95% confidence intervals. (**d**) As in (**b**), using phosphopeptide insulin response values. (**e**) Correlation of the Pfkfb2 S469 insulin response with insulin-stimulated ³H-2DG uptake. (**f**) Correlation of weighted gene correlation network analysis (WGCNA) subnetwork eigenpeptides with insulin-stimulated ³H-2DG uptake. Significant correlations are indicated (*). (**g**) Rationale and workflow for overexpressing Pfkfb3 to rescue palmitate-induced insulin resistance. (**h**) The fold change of unstimulated to insulin-stimulated glucose uptake (100 nM insulin, 20 min) in L6-GLUT4-HA myotubes with or without Pfkfb3 overexpression, treated with palmitate (125 μM, 16 hr) or BSA vehicle control. A two-way ANOVA was performed followed by Tukey's post hoc tests (*). Not all significant comparisons are shown. n=4 biological replicates. */#: 0.01≤p<0.05, **/##: 0.001≤p<0.01, ***/###: p<0.001.

The online version of this article includes the following source data and figure supplement(s) for figure 6:

**Figure supplement 1.** Characterisation of glucose uptake-correlated phosphosites.

**Figure supplement 2.** Overexpression of Pfkfb3 enhances glycolytic capacity and reverses palmitate-induced insulin resistance.

*Figure 6 continued on next page*

*Figure 6 continued*

**Figure supplement 2—source data 1.** Unedited scans of western blots for panels (b) and (f).

**Figure supplement 2—source data 2.** Pdf containing unedited scans of western blots alongside the edited scans in panels (b) and (f).

**Figure supplement 3.** Additional glucose uptake correlation analysis.

(*Figure 5f and h*), suggesting it may be a central regulatory hub for various outcomes of insulin action including glucose uptake. Examining the subnetwork membership scores for glucose uptake-correlated phosphopeptides also revealed a preference for clusters V and I, supporting this analysis (*Figure 6—figure supplement 1b–c*). Overall, compartmentalisation of insulin-responsive phospho-sites into subnetworks may enable independent control of insulin's functional outputs, since only two subnetworks correlated with insulin-stimulated glucose uptake.

## Upregulating glycolysis reverses insulin resistance

We next aimed to validate our approach for identifying regulatory mechanisms of insulin-stimulated glucose uptake. S469 on Pfkfb2 correlated highly with glucose uptake follow insulin stimulation (*Figure 6e*). Phosphorylation of this site leads to increased production of F2,6BP, a potent glycolytic agonist, suggesting that activating glycolysis may play a key role in muscle insulin responsiveness. This is consistent with our previous findings that glycolytic enzyme abundance was strongly associated with ex vivo insulin-stimulated glucose uptake in muscle from inbred mice (*Nelson et al., 2022*), that the insulin resistance-reversing small molecule thiostrepton enhances glycolytic capacity (*Masson et al., 2023*), and that decreasing glycolytic flux caused insulin resistance in vitro (*Trefely et al., 2015*). To further establish glycolysis as a regulator of insulin responsiveness in skeletal muscle, we decided to investigate whether upregulating glycolysis through F2,6BP production can restore insulin-stimulated glucose uptake in insulin resistance.

Since Pfkfb2 requires phosphorylation by Akt to produce F2,6BP substantially, increasing F2,6BP production via Pfkfb2 would require enhanced activating site phosphorylation, which is difficult to achieve in a targeted fashion, or phosphomimetic mutation of activating sites to aspartate/gluta-mate, which often does not recapitulate the molecular effects of serine/threonine phosphorylation (*Dephoure et al., 2013*). By contrast, the paralog Pfkfb3 has high basal production rates and lacks an Akt motif at the corresponding phosphosites (*Ros and Schulze, 2013*). We therefore rationalised that overexpressing Pfkfb3 would robustly increase F2,6BP production and enhance glycolysis regardless of insulin stimulation and Akt signalling (*Figure 6g*). To avoid systemic effects of Pfkfb3 overexpres-sion we studied cultured L6-GLUT4-HA myotubes, which display robust insulin regulation of GLUT4 trafficking and develop insulin resistance upon palmitate treatment, mimicking lipotoxicity, a trigger of in vivo insulin resistance (*Hoehn et al., 2009*). While L6 cells are of rat origin, they are preferable to the popular C2C12 mouse cell line since the latter lack an insulin-responsive vesicular compartment (*Tortorella and Pilch, 2002*) and undergo spontaneous contraction, resulting in confounding non-insulin-dependent glucose uptake (*Portiér et al., 1999*).

As anticipated, Pfkfb3 overexpression increased glycolytic capacity in L6-GLUT4-HA myotubes as measured by extracellular acidification rate (ECAR) (*Figure 6—figure supplement 2a–c*). Pfkfb3 over-expression also restored insulin-responsive glucose uptake to normal levels in palmitate-treated cells (*Figure 6h*; *Figure 6—figure supplement 2d*). This effect was only observed in cells treated with palmitate and insulin, suggesting it specifically modulated insulin action rather than non-specifically increasing glucose uptake through enhanced glucose consumption (*Figure 6—figure supplement 2d*). This may be due to enhanced insulin signalling, as we have previously observed that transient Pfkfb3 overexpression increased Akt signalling in HEK293 cells (*Trefely et al., 2015*). However, immunoblotting of canonical insulin-responsive phosphosites on Akt and its substrates GSK3α/β and PRAS40 revealed minimal effect of palmitate treatment and Pfkfb3 overexpression (*Figure 6—figure supplement 2e–f*), hence more detailed phosphoproteomics studies are needed to clarify whether Pfkfb3 overexpression restored insulin action by modulating insulin signalling. Overall, our results further establish glycolytic flux as a major determinant of the glucose uptake arm of muscle insulin action and highlight the power of studying phosphoproteomics across the gene-by-environment land-scape to identify causal drivers of complex phenotypes. We anticipate that our catalogue of glucose

uptake-correlated phosphosites will provide a rich starting point for future investigations into mechanisms of insulin action and resistance.

## Discussion

The environment shapes the flow of information from genotype to phenotype. Many studies have interrogated the role of intermediate molecular layers such as the transcriptome or proteome, however few studies have examined how protein post-translation modifications participate in this information transfer. Here, we have approached this problem by leveraging diverse inbred mouse strains and phosphoproteomics to examine the insulin signalling network across a landscape of genetic and dietary variation. Genetic background significantly altered the insulin signalling network both independently and in concert with dietary status, affecting myriad phosphosites and multiple kinases. We exploited this variation in signalling responses in two ways – to study the partitioning of the Akt and insulin signalling pathways into distinct subnetworks of coregulated phosphosites; and to identify potential regulators of insulin responsiveness by associating phosphorylation with insulin-stimulated glucose uptake. Finally, validation studies in L6 myotubes confirmed the major role of accelerated glycolysis as a key regulator of insulin responsiveness.

Genetic and diet-driven signalling changes did not transmit linearly through our current model of the insulin signalling network, illustrating that this model remains incomplete. Notably, substrates of kinases such as Akt clustered into distinct groups based on differing insulin responses. Hence, it is an oversimplification to model signalling pathways as networks of individual kinases since substrates of the same kinase display independent regulation. This could arise from localisation of a kinase to distinct substrate pools (*Bloom and Cross, 2007*; *Steinberg and Hardie, 2023*; *Lavoie et al., 2020*); interactors targeting a kinase to different substrates (*Bloom and Cross, 2007*); substrate phosphorylation by alternate kinases (*Meyuhas, 2015*); the dephosphorylation of specific substrates by phosphatases; kinase post-translational modifications altering substrate specificity; and distinct substrate phosphorylation kinetics (*Humphrey et al., 2013*). As our knowledge of the repertoire of kinase substrates continues to deepen (*Johnson et al., 2023*), future research should explore how the above mechanisms contribute to finer regulation of these substrates. Genetic and environmental variation also exposed a coregulation subnetwork structure within the insulin signalling network. The enrichment of subnetworks in distinct biological processes, and the selective association of two subnetworks with glucose uptake, suggests that this coregulation structure may direct independent control of distinct outcomes of insulin action. This exciting possibility necessitates further investigation, including replication in independent cohorts and spatiotemporal characterisation of subnetwork dynamics. Alternatively, one could overlap subnetworks with genetic information, such as genes associated with glucose homeostasis and other metabolic traits in human GWAS (*Chen et al., 2021*), or muscle-specific eQTLs or pQTLs genetically colocalised with similar traits (*Molendijk et al., 2022*), to further prioritise subnetwork-associated phenotypes and identify potential drivers within subnetworks.

Muscle insulin signalling responses vary across individuals (*Needham et al., 2022*; *Tonks et al., 2013*), and our results suggest that baseline genetic differences and an individual's environment both alter signalling, with the environment's influence depending strongly on genetic background. Signalling pathways are popular therapeutic targets due to their importance in human health and the relative ease of pharmaceutical interventions (*Attwood et al., 2021*). Our results advocate for a personalised approach to such therapies, implying that the efficacy of treatments aiming to correct pathological signalling responses will depend on an individual's genetic background. In cancer, where signalling networks are dysregulated heterogeneously, modelling patient-specific networks has already shown promise for identifying personalised drug targets (*Eduati et al., 2020*; *Montagud et al., 2022*). Personalised medicine approaches will also be aided by a comprehensive understanding of how genetics shape signalling networks and potentiate their modulation by the environment. Recent studies have made the first step, revealing that the ground-state phosphoproteome can be altered by mutations affecting network components such as kinases, phosphatases, and phosphoproteins, as well as the molecular milieu the network is exposed to including extracellular signalling ligands (*Grossbach et al., 2022*; *Zhang et al., 2023*). An important corollary of such genetic factors is that multiple genetic backgrounds should be studied when establishing generalisable signalling responses. Our data indicate that insulin responses in C57BL6J – the most commonly studied mouse

strain – are not necessarily generalisable, indicated by phosphosites such as S15 on Rnmt that were insulin regulated almost exclusively in C57Bl6J.

A major challenge in studying signal transduction with omics technologies is that hundreds to thousands of molecular events typically respond to a cellular signal, making it difficult to pinpoint the most crucial regulatory nodes. To tackle this challenge, we previously demonstrated that associating phosphoproteomics with physiological phenotypes across diverse individuals enriches for phosphosites more likely to modulate biological responses (*Needham et al., 2022*). Here, we have elaborated on this approach, revealing that associating phosphorylation with phenotype across a genetic and environmental landscape can identify regulators of specific biological processes, such as insulin-stimulated glucose uptake. Our results recapitulated known glucose uptake regulators and led to further validation of glycolytic flux as a modulator of insulin responsiveness. We have previously demonstrated that reduced glycolytic flux impairs GLUT4 translocation and insulin signalling (*Trefely et al., 2015*), implying that the status of glycolysis is sensed by proteins regulating insulin action. An enhanced glycolytic metabolic tone may alter production of reactive oxygen species, a known modulator of insulin action (*Hoehn et al., 2009*; *Houstis et al., 2006*; *Fazakerley et al., 2018*) and insulin signalling (*Su et al., 2019*). Alternatively, recent approaches to map protein-metabolite interactions could identify points of allosteric crosstalk between glycolytic metabolites and insulin action proteins (*Piazza et al., 2018*; *Hicks et al., 2023*), hence broadening our understanding of the bidirectional communication between insulin action and metabolism.

It was striking that only several dozen insulin-regulated phosphopeptides associated significantly with glucose uptake. Since insulin triggers multiple distinct cellular outcomes, it is possible that only a subset of insulin-responsive phosphosites contribute to enhanced glucose uptake. Moreover, many of these phosphosites might only be permissive for insulin-stimulated glucose uptake and are not major regulatory nodes determining the fidelity of the process. For example, while mutation of the four primary Akt regulatory sites on Tbc1d4 blocks GLUT4 translocation (*Sano et al., 2003*), none of these phosphosites featured strong positive correlations with glucose uptake in our analysis (*Figure 6—figure supplement 3*). This implies that their phosphorylation may promote glucose uptake in a functionally permissive, switch-like fashion. We suggest that the glucose uptake-associated phosphosites we have identified will be enriched in major regulators of insulin responsiveness, necessitating future functional studies to characterise these sites and explore their involvement in insulin resistance.

Our work demonstrates that genetic and environmental variation can profoundly modulate global signalling networks and that the influences of these factors are intrinsically entwined. We show that the resulting diversity in signalling responses can be leveraged to pinpoint regulators of insulin-stimulated glucose uptake, providing a powerful methodological framework for interrogating the regulatory basis of complex biological pathways.

## Limitations of this study

First, this study focused on male mice and examined only five inbred strains. This limited number of strains may mean that our association analysis was underpowered to detect some regulators of insulin responsiveness. Importantly, however, this does not imply that the regulators identified are incorrect, but only that there may be more to discover with larger cohorts. In addition, studies incorporating male and female subjects found that mouse insulin sensitivity was affected by interactions between sex and strain (*Parks et al., 2015*), and insulin signalling in stem cell-derived myoblasts by interactions between sex and donor metabolic health (*Haider et al., 2021*). Hence, we anticipate that sex would contribute another important dimension to the web of interactions between genetics, diet, and insulin signalling identified here. Future work should therefore extend our approach across a broader range of genetic backgrounds, as well as in female mice. Second, we only examined insulin signalling after 10 min, since measuring multiple timepoints would have drastically increased the number of animals and samples required. Integration of dynamic phosphoproteome data from cultured cells indicated that insulin signalling dynamics may contribute to trends in our data (*Figure 5i*), suggesting the exploration of signalling at additional timepoints may be fruitful in the future.

Third, mammalian tissues are a heterogeneous mixture of cell types, and differences in this mixture could result in different signalling responses measured at the whole tissue level. In our experience, the soleus can be reproducibly dissected as an intact muscle with little contamination from surrounding tissues, making it unlikely that cell-type composition varied across samples due to tissue collection.

However, we cannot exclude the possibility that differences in the composition of the soleus muscle across strains and diets contributed to the signalling changes we detected. We were also unable to determine the extent to which signalling changes arose from muscle-intrinsic or -extrinsic factors. For instance, body weight varied substantially across mice and correlated significantly with 25% of Strain- and Diet-affected phosphopeptides (*Figure 6—figure supplement 3c*), suggesting obesity-related systemic factors likely impact a subset of the muscle insulin signalling network. Furthermore, genetic differences in lifespan could alter the 'biological age' of different strains and their phosphoproteomes, though we could not assess this possibility since lifespan data is not available for most strains used. Lastly, as we did not perform total proteomics, we did not assess whether phosphosite changes were caused by differences in total protein abundance. However, since protein abundance should not change within a 10 min insulin stimulation (*Fazakerley et al., 2023*), the acute insulin responses we observed should only reflect differences in protein phosphorylation. Furthermore, in our previous studies of insulin signalling in adipocytes (*Fazakerley et al., 2023*) or human skeletal muscle (*Needham et al., 2022*) in which deep proteomes were measured in parallel, we found little global correlation between changes in protein phosphorylation and protein abundance across experimental conditions regardless of insulin stimulation, suggesting the contribution of protein abundance to phosphosite changes across strains and diets was likely minimal.

## Methods
### Statistical analysis
Most statistical analysis was performed in the R programming environment using RStudio (R version: 4.2.1, RStudio version: 2022.07.1 Build 554). Analysis of GLUT4-HA-L6 myotube Pfkfb3 expression, 2DG uptake, and ECAR was performed in GraphPad Prism (version: 9.3.1).

### Animal details
Male C57BL/6J (C57Bl6J), BXH9/TyJ (BXH9), BXD34/TyJ (BXD34), and CAST/EiJ (CAST) mice were purchased from Australian BioResources (Moss Vale, NSW, Australia) while NOD/ShiLtJ (NOD) mice were purchased from Animal Resources Centre (Murdoch, WA, Australia). Mice were at most 9 weeks of age upon arrival. Mice were housed at 23°C on a 12 hr light/dark cycle in cages of 2–5, with free access to food and water. At 12–16 weeks of age mice were randomly allocated to a standard CHOW diet (13% calories from fat, 65% from carbohydrate, 22% from protein; 'Irradiated Rat and Mouse Diet', Specialty Feeds, Glen Forrest, WA, Australia) or a high-fat high-sucrose diet made in house (HFD; 45% calories from fat [40% calories from lard], 35% from carbohydrate [14% calories from starch], and 22% from protein) and sacrificed exactly 6 weeks later. The number of mice in each group are C57Bl6J: 8 CHOW, 10 HFD; NOD: 10 CHOW, 10 HFD; BXH9: 8 CHOW, 9 HFD; CAST: 9 CHOW, 9 HFD; BXD34: 10 CHOW, 11 HFD. Procedures were carried out with approval from the University of Sydney Animal Ethics Committee following guidelines issued by NHMRC (Australia).

### Assessment of body composition
Body composition of individual mice was assessed using the EchoMRI-900 to determine lean mass 1 day before a glucose tolerance test and 5–6 days before euthanasia. Analysis was performed as per the manufacturer's specifications.

### Glucose tolerance test
On the day of a glucose tolerance test mice were fasted for 6 hr (0800–1400). Mice were then orally gavaged with 20% (wt/vol) glucose in water at 2 g/kg lean mass, and blood glucose was measured from the tail vein using a glucometer 0, 15, 30, 45, 60, and 90 min after the gavage. At 0 and 15 min, 5 µL blood was also collected into an Insulin Mouse Ultra-Sensitive ELISA plate (Crystal Chem USA, Elk Grove Village, IL, USA). Blood insulin concentration was measured according to the manufacturer's protocol, using linear extrapolation from an insulin standard curve. The area of the blood glucose curve (AOC) was calculated by:

$$AOC = \sum_{i=2}^{n} \frac{(G_{i-1} - G_1) + (G_i - G_1)}{2}(t_i - t_{i-1})$$

where $i$ represents the $i$th timepoint at which blood glucose was measured, $n$ represents the last timepoint, $t_i$ represents the time (min) of the $i$th timepoint, and $G_i$ represents blood glucose concentration (mM) at the $i$th timepoint.

## In vivo insulin stimulation

On the day of the procedure mice were fasted for 2 hr (1100–1300). Mice were then anaesthetised by intraperitoneal injection of sodium pentobarbital at 80 mg/kg body mass. To counter anaesthesia-associated declines in body temperature, mice were wrapped in aluminium foil and placed on a heating pad at 37°C. After 15 min anaesthesia, mice were injected retro-orbitally as previously described (*Yardeni et al., 2011*) with 50 µL plasma replacement (B. Braun, Melsungen, DEU) containing 10 µCi [3]H-2DG and saline or insulin (0.75 U/kg lean mass). Blood glucose was measured from the tail vein using a glucometer (AccuCheck, Roche Diabetes Care, NSW, Australia) 1 min prior to injection and 1, 5, 7.5, and 10 min after injection. Simultaneously, 5 µL blood was collected into 95 µL 0.9% NaCl on ice to measure [3]H-2DG blood concentration. Ten minutes after insulin injection mice were sacrificed by cervical dislocation and the soleus muscle was rapidly excised, immediately frozen in liquid nitrogen, and stored at –80°C. To measure [3]H-2DG blood concentration, diluted blood samples were first centrifuged at 10,000 × $g$ for 10 min to pellet blood cells. Supernatant (70 µL) was collected and combined with 3 mL liquid scintillation cocktail (Perkin Elmer, MA, USA: 6013321) to allow the measurement of [3]H with a Tri-Carb 2810TR Liquid Scintillation Counter (Perkin Elmer, MA, USA).

## Skeletal muscle lysis

Frozen muscle tissue was powdered by grinding in a mortar and pestle chilled with liquid nitrogen and dry ice. To lyse powdered tissue, 200 µL lysis buffer (4% [wt/vol] sodium deoxycholate, 100 mM Tris-HCl pH 8.5) was added followed by 10 s vortexing. Samples were then sonicated at 4°C at 90% power using pulses of 2 s on, 5 s off for a total time of 1 min. Samples were then immediately boiled at 95°C with 1500 rpm shaking for 5 min and sonicated for a further 2 min (4°C, 90% power, 5 s on and 5 s off) to ensure complete lysis. Lysate was then centrifuged at 20,000 × $g$ for 5 min and 180 µL supernatant was collected. Cysteine residues were reduced and alkylated by adding 40 mM chloroacetamide and 10 mM tris 2-carboxyethyl phosphine (TCEP) at pH 7. Lysate was incubated for 5 min at 45°C with 1500 rpm shaking and then incubated for a further 40 min at room temperature without shaking.

Next, 800 µL 100% chloroform and 1600 µL 100% methanol were added following 30 s sonication at 90% power. LC/MS grade water (800 µL) was added following 5 min mixing at 1000 rpm. Lysate was centrifuged for 5 min at 2000 × $g$ to induce a phase separation. The majority of the aqueous phase (2400 µL) was removed and 2000 µL was reserved for [3]H-2DG quantification. Next, 2400 µL 100% methanol was added following 30 s mixing at 800 rpm and centrifugation at 2000 × $g$ for 5 min. The supernatant was discarded, and the protein pellet was air-dried for 5 min. Protein was reconstituted in 200 µL lysis buffer, sonicated at 60% power for 15 s using a tip-probe sonicator, and boiled for 5 min in a thermomixer at 95°C with 1500 rpm shaking.

## Determining [3]H-2DG uptake into muscle tissue

Anion exchange chromatography was used to quantify [3]H-p2DG, representing [3]H-2DG that has been taken up by cells. For quantification of total (phosphorylated and unphosphorylated) [3]H-2DG, 375 µL lysate aqueous phase was combined with 1125 µL water and reserved. For quantification of unphosphorylated [3]H-2DG, 1000 µL lysate aqueous phase was added to 300 µL 37.5% (wt/vol) AG 1-X8 anion exchange resin (Bio-Rad, Hercules, CA, USA: 1401441) and washed with 3 mL water to remove p2DG. Liquid scintillation cocktail (3 mL) was then added to 1500 µL total and unphosphorylated [3]H-2DG solutions, and [3]H-2DG was measured using a Tri-Carb 2810R Liquid Scintillation Counter. Unphosphorylated and total [3]H-2DG scintillation counts were subtracted to quantify [3]H-p2DG.

[3]H-2DG blood concentration at 1, 5, 7.5, and 10 min after injection was fit to an exponential curve $y = C_p(0) e^{-K_p t}$, where $C_p(0)$ represents the predicted initial tracer concentration (DPM/µL) and $K_p$ represents the rate of tracer disappearance from the blood (1/min), to model the disappearance of [3]H-2DG from the blood as it is taken up and trapped by peripheral tissues (*Goodner et al., 1980*). $C_p(1)$ was removed when it was abnormally low ($C_p(1) < C_p(5), C_p(5) - C_p(1) > 0.5 \times (C_p(5) - C_p(7.5))$), which likely indicates insufficient diffusion of circulating [3]H-2DG into the tail vein. To account for different

rates of blood $^3$H-2DG disappearance across mice, $^3$H-2DG uptake was calculated as a rate constant (*Goodner et al., 1980*):

$$K_i = \frac{C_i\left(t\right) K_p}{C_p\left(0\right)\left(1 - e^{-K_p t}\right)}$$

where $t$ represents the time after injection that the animal was sacrificed (min) and $C_i\left(t\right)$ represents the concentration of $^3$H-p2DG in the tissue harvested at time $t$ (DPM/mg tissue).

## Phosphoproteomics sample preparation

Phosphopeptides were isolated using the EasyPhos protocol (*Humphrey et al., 2018*) with minor modifications. Protein (C57Bl6J and NOD: 755 µg, BXH9 and BXD34: 511 µg, CAST: 364 µg) was digested into peptides by incubation in 1% (wt/wt) Trypsin and LysC on a thermomixer at 37°C with 1500 rpm shaking for 14 hr. Following digestion, 400 µL 100% isopropanol and 100 µL EasyPhos enrichment buffer (48% [vol/vol] TFA, 8 mM KH$_2$PO$_4$) were sequentially added with mixing (1500 rpm, 30 s) after each addition. Lysate was centrifuged at 20,000 × $g$ for 15 min to pellet insoluble material and transferred to a deep well plate. The EasyPhos protocol was then followed from step 12 (*Humphrey et al., 2018*).

## Liquid chromatography-tandem mass spectrometry

Enriched phosphopeptides in MS loading buffer (2% ACN, 0.3% TFA) were loaded onto in-house fabricated 55 cm columns (75 µM ID), packed with 1.9 µM C18 ReproSil Pur AQ particles (Dr. Maisch HPLC GmbH, Ammerbuch, DEU) with a Dionex U3000 HPLC (Thermo Fisher Scientific), interfaced with an Orbitrap Exploris 480 mass spectrometer (Thermo Fisher Scientific). Column temperature was maintained at 60°C using a column oven (Sonation lab solutions, Biberach, DEU), and peptides were separated using a binary buffer system comprising 0.1% formic acid (buffer A) and 80% ACN plus 0.1% formic acid (buffer B) at a flow rate of 400 nL/min. A gradient of 3–19% buffer B was employed over 40 min followed by 19–41% buffer B over 20 min, resulting in approximately 1 hr gradients. Peptides were analysed with one full scan (350–1400 m/z, $R$=120,000) at a target of 3e$^6$ ions, followed by 48 DIA MS/MS scans (350–1022 m/z) with higher-energy collisional dissociation (target 3e$^6$ ions, max injection time 22 ms, isolation window 14 m/z, 1 m/z window overlap, normalised collision energy 25%), and fragments were detected in the Orbitrap ($R$=15,000).

## MS raw data processing

Raw spectral data were analysed using Spectronaut (v16.0.220606.53000). Data were searched using directDIA against the Mouse UniProt Reference Proteome database (January 2022 release), with default settings (precursor and protein Qvalue cutoffs 0.01, Qvalue filtering, MS2 quantification), with 'PTM localisation' filtering turned on (threshold 0.5), and the inbuilt peptide collapse function.

## Phosphoproteomics data processing

Phosphopeptide intensities were log2 transformed and median normalised. Non-class I phosphopeptides (maximum localisation score across samples ≤0.75) were then removed. Finally, for each phosphopeptide, outlier values were removed that had a log2 intensity <5 and were >6 log2 intensity units lower than the phosphopeptide median. Log2 fold changes between conditions were computed using condition medians.

## Identifying insulin-regulated phosphopeptides

To allow comparison across conditions, phosphopeptides were filtered for those highly quantified in most strain-diet combinations. For a given phosphopeptide, this filtering was performed on two levels. Firstly, each of the 10 strain-diet combinations were retained if there were ≥3 insulin-stimulated values and ≥3 unstimulated values. Then, the phosphopeptide itself was retained if ≥8 strain-diet combinations had passed the previous filtering. Phosphopeptides were then fit to a three-way ANOVA with all interaction terms ('aov' in the R package 'stats') and an F-test was performed assessing the main effect of insulin stimulation. To correct for multiple hypothesis testing p-values were converted into q-values (R package 'qvalue'; *Storey, 2002*). The log2(insulin/unstimulated) fold change with the greatest magnitude across strain-diet combinations was then calculated (max log2(insulin/

unstimulated)). Phosphopeptides were considered 'insulin-regulated' when q<0.05 and if insulin increased or decreased phosphorylation by >1.5-fold in at least one strain-diet combination (max log2(insulin/unstimulated)>0.58 or <–0.58).

## Calculation of insulin response values

For all phosphopeptides the distribution of 'insulin responses' in each strain-diet combination was calculated. Specifically, within each strain-diet combination all insulin-stimulated values were normalised by subtracting the unstimulated median.

## Identifying strain and diet effects

### Strain effects

For each insulin-regulated phosphopeptide a one-way ANOVA was performed modelling the insulin response as a function of mouse strain within the CHOW diet. p-Values were converted to q-values. For significant phosphopeptides (q<0.05), t-tests were performed comparing the insulin response of C57Bl6J to each of the other four strains. t-Test p-values were converted to q-values and considered significant when q<0.05. To ensure that significant differences between a strain and C57Bl6J were of a meaningful magnitude, the strain's log2(insulin/unstimulated) was compared to the C57Bl6J log2(-insulin/unstimulated). In general, if the absolute difference between the two was greater than 0.58 this was accepted. However, this threshold was decreased for phosphopeptides with weaker insulin regulation. Specifically, the difference was retained if it passed the following filtering:

$$Strain\ log2(insulin/unstimulated) > f(C57Bl6J\ log2(insulin/unstimulated))$$

or

$$Strain\ log2(insulin/unstimulated) < g(C57Bl6J\ log2(insulin/unstimulated))$$

where $f(x)$ and $g(x)$ are defined as:

$$f(x) = \begin{cases} x + 0.58, & if\ x \geq 2 \times 0.58 \\ \frac{5}{4}x + \frac{1}{2} \times 0.58, & if\ 0 \leq x < 2 \times 0.58 \\ x + \frac{1}{2} \times 0.58, & if -\frac{1}{2} \times 0.58 \leq x < 0 \\ \frac{4}{5}x + \frac{2}{5} \times 0.58, & if -3 \times 0.58 \leq x < -\frac{1}{2} \times 0.58 \\ x + 0.58, & if\ x < -3 \times 0.58 \end{cases}$$

$$g(x) = \begin{cases} x - 0.58, & if\ x \geq 3 \times 0.58 \\ \frac{4}{5}x + \frac{2}{5} \times 0.58, & if\ \frac{1}{2} \times 0.58 \leq x < 3 \times 0.58 \\ x - \frac{1}{2} \times 0.58, & if\ 0 \leq x < \frac{1}{2} \times 0.58 \\ \frac{5}{4}x - \frac{1}{2} \times 0.58, & if -2 \times 0.58 \leq x < 0 \\ x - 0.58, & if\ x < -2 \times 0.58 \end{cases}$$

An insulin-regulated phosphopeptide was considered to have a 'Strain effect' if the insulin response in at least one strain was different to C57Bl6J using the q-value and log2 fold change criteria described above.

### Uniform diet and Strain×Diet effects

For each insulin-regulated phosphopeptide a two-way ANOVA was performed modelling the insulin response as a function of strain, diet, and their interaction. The p-values for the Diet and Strain×Diet terms were converted to q-values. Whenever the Strain×Diet term was significant (q<0.05), additional tests were performed to identify specific strains in which the insulin response differed between CHOW and HFD. If the Strain×Diet term was not significant but the Diet term was significant, a separate filtering procedure was performed.

When the Strain×Diet term was significant, t-tests were performed to compare the CHOW insulin response to the HFD insulin response within each strain. When a t-test was significant (q<0.05), the

log2(insulin/unstimulated) filtering procedure described for 'Strain effects' was applied comparing CHOW and HFD fold changes. Insulin-regulated phosphopeptides were considered to have a 'Strain×Diet effect' if there was a difference between CHOW and HFD in at least one strain.

When only the Diet term was significant, the log2(insulin/unstimulated) filtering procedure described for 'Strain effects' was applied, comparing the mean log2(insulin/unstimulated) across strains within CHOW, to the mean across HFD. Insulin-regulated phosphopeptides that passed this filter were considered to have a 'Uniform diet effect'.

## Curated insulin signalling subnetwork

A subnetwork of insulin-regulated phosphosites was curated by compiling all sites on proteins from a previously published knowledge pathway-derived list of canonical insulin signalling proteins (*Needham et al., 2022*). Several phosphosites and proteins that were not detected as insulin regulated were included due to their importance in the insulin signalling pathway. Additionally, all in vivo substrates of canonical insulin-regulated kinases (Akt, mTOR, AMPK, Raf, Mek1/2, Erk1/2, p90RSK/ Rsk2, p70S6K, Pdk1, INSR) annotated in PhosphositePlus were included (*Hornbeck et al., 2015*). Annotations from orthologous phosphosites were pooled across species using PhosphositePlus Site Group IDs. Phosphosite regulatory roles from PhosphositePlus were indicated after manual validation by literature search. Proteins were assigned to functional groups (e.g. mRNA processing, lipid metabolism) based on their Uniprot descriptions.

## Kinase substrate enrichment analysis

Kinase substrate annotations were collated from PhosphositePlus and mapped into phosphoproteomics data using Site Group IDs. Only annotations supported by in vivo evidence were used. Annotations for kinase isoforms (e.g. Akt1, Akt2, Akt3) were merged. Substrate annotations for GSK3 were supplemented with a recent list of 274 putative GSK3 substrates determined by phosphoproteomics and motif analysis (*Fazakerley et al., 2023*). Autophosphorylation sites and promiscuous phosphosites targeted by ≥4 kinases were removed. KSEA was then performed with the 'ksea' function from the R package 'ksea' (*Hernandez-Armenta et al., 2017*) (version: 0.1.2) using insulin response data and 1000 permutations to determine empirical p-values. Only phosphopeptides quantified in ≥50% of samples and with ≥1 insulin response value in all strain-diet combinations were used. In each sample kinases with <5 quantified substrates were excluded, and only kinases with significant enrichment ($p < 0.05$) in ≥5 samples were used in subsequent analysis. To identify Strain effects on kinase activity, one-way ANOVAs were performed on CHOW KSEA enrichment scores. To identify Uniform diet or Strain×Diet effects, two-way ANOVAs were performed on KSEA enrichment scores testing the effects of strain, diet, and their interaction. p-Values were adjusted by the Benjamini-Hochberg procedure.

## Insulin signalling subnetwork analysis

WGCNA (*Zhang and Horvath, 2005*; *Langfelder and Horvath, 2008*) was performed with the 'block-wiseModules' function from the R package 'WGCNA' (version 1.71) using the insulin response values of all insulin-regulated phosphopeptides. Default parameters were used except for power = 3 (determined as recommended in *Zhang and Horvath, 2005*), deepSplit = 3, minModuleSize = 15, reassignThreshold = 0, and mergeCutHeight = 0.25. Subnetwork eigengenes were extracted and termed 'eigenpeptides'.

One-sided Fisher's exact tests were performed to assess the enrichment of GO Biological Processes, GO Cellular Compartments (R package 'org.Mm.eg.db' version 3.15.0) (*Carlson, 2019*), and kinase substrates in each subnetwork relative to the entire phosphoproteome. Only pathways containing three or more subnetwork phosphoproteins were tested. Kinase substrate enrichment was performed using the same annotations as KSEA. p-Values were adjusted within each analysis by the Benjamini-Hochberg procedure. Subnetwork phosphopeptides were mapped into insulin signalling temporal clusters defined in our previous study of insulin signalling dynamics (*Humphrey et al., 2013*), using PhosphositePlus Site Group IDs. The timepoint at which each cluster appeared to reach its maximum insulin-stimulated value was used as a measure of insulin response speed.

## Glucose uptake correlations

For each insulin-regulated phosphopeptide, Pearson's correlation tests were performed to assess the linear association between $^3$H-2DG uptake in insulin-stimulated mice and phosphopeptide insulin response values or unnormalised insulin-stimulated log2 intensity. Phosphopeptides were considered correlated with $^3$H-2DG uptake when q<0.1 and their Pearson's correlation coefficient was of substantial magnitude ($r$>0.35 or $r$<–0.35). Pearson's correlation tests were also performed comparing insulin-stimulated $^3$H-2DG uptake to KSEA enrichment scores in individual mice or using the median in each strain-diet combination.

## Cell lines

GLUT4-HA-L6 myoblasts (*Carey et al., 2006*) were grown in α-MEM supplemented with 10% fetal bovine serum in a humidified chamber at 37°C, 10% $CO_2$. Differentiation was induced by changing media to α-MEM supplemented with 2% horse serum for 5 days. Cells were frequently tested for mycoplasma which always revealed no contamination.

## Pfkfb3 overexpression

Platinum-E (Plat-E) retroviral packaging cells were grown to confluency and transfected with 10 μg total DNA: either pBabe puromycin empty vector, pBabe puromycin Pfkfb3, or pWZL neomycin HA-GLUT4. After 48 hr retroviral media was collected and passed through a 0.45 μm filter. L6 myotubes were grown to confluence and retrovirally transfected with 2 mL of HA-GLUT4 neomycin viral media in the presence of 10 μg/mL polybrene. The following morning, cells were split into growth media containing neomycin (800 μg/mL) and passaged until only successfully transfected cells remained. These cells were then grown to confluence and retrovirally transfected again with 2 mL of either empty vector puromycin viral media or Pfkfb3 puromycin viral media in the presence of 10 μg/mL of polybrene. The following morning, cells were split into growth media containing both neomycin (800 μg/mL) and puromycin (2 μg/mL) and passaged until only successfully transfected cells remained in culture.

## Extracellular acidification rate

The ECAR of GLUT4-HA-L6 cells myotubes was measured using Seahorse XFp miniplates and a Seahorse XF HS Mini Analyzer (Seahorse Bioscience, Copenhagen, Denmark) as previously described (*Krycer et al., 2020*). Cells incubated in palmitate or BSA control were washed twice with Krebs-Ringer Bicarbonate Buffer (Sigma, K4002) and once with standard cell culture media without bicarbonate (XF-DMEM, pH 7.4). Cells were then incubated in XF-DMEM without glucose at 37°C for 1 hr in a non-$CO_2$ incubator, followed by assaying in the XFp Analyzer. ECAR was measured after a 12 min equilibration period followed by mix/wait/read cycles of 3/0/3 min. After stabilising the baseline rates, compounds were injected to reach a final concentration of: 10 mM glucose, 5 μg/mL oligomycin, and 50 mM 2-deoxyglucose (2DG), allowing estimation of glucose-driven glycolysis (glucose ECAR – basal ECAR), glycolytic capacity (oligomycin ECAR – 2DG ECAR), and non-glycolytic acidification (equal to 2DG ECAR). Data were normalised to protein concentration and presented as a percentage of total ECAR.

## 2DG uptake in GLUT4-HA-L6 myotubes

2DG uptake into GLUT4-HA-L6 myotubes was performed as previously described with modifications (*Carey et al., 2006*; *Masson et al., 2021*). Cells were incubated overnight in α-MEM supplemented with either BSA-coupled 125 μM palmitic acid or BSA vehicle control before being washed 3× with 37°C HEPES-buffered saline (HBS). Cells were then incubated in HBS supplemented with 10 μM unlabelled 2DG and either 0 or 100 nM insulin at 37°C for 15 min. Cells were then incubated for a further 5 min following the addition of 0.5 μCi/ml [$^3$H]-2DG in HBS. Cells were then washed on ice 5× with ice-cold PBS and lysed in 1 M NaOH. For non-specific background uptake, one well per condition was pre-treated with cytochalasin B. Counts were determined by Perkin Elmer Quantulus GCT Liquid Scintillation Counter (Perkin Elmer, Waltham, MA, USA). 2DG uptake was expressed relatively to protein concentration as determined by bicinchoninic acid assay after neutralisation with 1 M HCl and subtraction of non-specific uptake.

## Immunoblotting

GLUT4-HA-L6 myotubes were incubated overnight (16 hr) in either BSA-conjugated 125 µM palmitate or BSA vehicle control. Cells were optionally treated with insulin as described above, and then washed in ice-cold PBS and lysed by scraping directly into 55°C Laemmli sample buffer with 10% (TCEP). Samples were sonicated for 24 s (3 s on/3 s off) and heated at 65°C for 5 min. Samples were then resolved by SDS-PAGE as previously described (*Nelson et al., 2022*), transferred onto PVDF membranes and blocked in TBS-T (0.1% Tween in Tris-buffered saline) containing 5% skim milk for 1 hr. Membranes were then washed 3×10 min in TBS-T and incubated overnight in primary antibodies against Pfkfb3 (Proteintech Group; 13763-1-AP) and α-tubulin (Cell Signalling Technologies #2125; diluted 1:1000). Insulin-stimulated cells were additionally incubated in primary antibodies against pT308 Akt (Cell Signaling Technologies #2965; diluted 1:1000), pS473 Akt (Cell Signaling Technologies #9271; diluted 1:1000), total pan-Akt (Cell Signaling Technologies #9272; diluted 1:1000), pS21/S9 GSK3α/β (Cell Signaling Technologies #9327; diluted 1:1000), pT246 PRAS40 (Cell Signaling Technologies #13175; diluted 1:1000), total PRAS40 (Cell Signaling Technologies #2691, diluted 1:1000), and 14-3-3 (Santa Cruz #sc-1657, diluted 1:5000). The following day membranes were washed 3×10 min in TBS-T and incubated for 1 hr in species-appropriate fluorescent antibodies. Imaging and densitometry were performed using LI-COR Image Studio.

## Acknowledgements

DEJ is an Australian Research Council (ARC) Laureate Fellow. The content is solely the responsibility of the authors and does not necessarily represent the official views of the ARC. The authors also acknowledge the facilities, and the scientific and technical assistance of the Sydney Mass Spectrometry Facility and the Laboratory Animal Services at the Charles Perkins Centre, University of Sydney.

## Additional information

### Funding

| Funder | Grant reference number | Author |
|---|---|---|
| Australian Research Council | ARC Laureate Fellowship | David E James |

The funders had no role in study design, data collection and interpretation, or the decision to submit the work for publication.

### Author contributions

Julian van Gerwen, Conceptualization, Formal analysis, Investigation, Methodology, Project administration, Visualization, Writing – original draft, Writing – review and editing; Stewart WC Masson, Sean J Humphrey, Investigation, Supervision, Writing – review and editing, Formal analysis, Writing – original draft, Project administration; Harry B Cutler, Alexis Diaz Vegas, Meg Potter, Writing – review and editing, Formal analysis, Project administration; Jacqueline Stöckli, Resources, Writing – review and editing, Project administration; Søren Madsen, Marin E Nelson, Writing – review and editing, Project administration; David E James, Investigation, Supervision, Funding acquisition, Writing – original draft, Project administration

### Author ORCIDs

Julian van Gerwen http://orcid.org/0000-0002-8093-0704
Stewart WC Masson http://orcid.org/0000-0003-4514-7009
Harry B Cutler https://orcid.org/0000-0002-2074-8599
Sean J Humphrey https://orcid.org/0000-0002-2666-9744
David E James http://orcid.org/0000-0001-5946-5257

### Ethics

Procedures were carried out with approval from the University of Sydney Animal Ethics Committee following guidelines issued by NHMRC (Australia; protocol 1988).

Reviewer #1 (Public Review): https://doi.org/10.7554/eLife.89212.3.sa1
Reviewer #2 (Public Review): https://doi.org/10.7554/eLife.89212.3.sa2
Reviewer #3 (Public Review): https://doi.org/10.7554/eLife.89212.3.sa3
Author Response https://doi.org/10.7554/eLife.89212.3.sa4

## Additional files

### Supplementary files

• Supplementary file 1. Muscle phosphoproteomics. (Page 1 '01_quantification') Normalised LFQ intensities of class I phosphopeptides. (Page 2 '02_analysis') Statistical analysis of phosphoproteome data.

• Supplementary file 2. Overlap between insulin and exercise-regulated phosphosites. Insulin-regulated phosphosites from this study that were additionally regulated by exercise either in *Needham et al., 2022* or *Hoffman et al., 2015*.

• Supplementary file 3. Insulin signalling subnetworks. Weighted gene correlation network analysis (WGCNA)-derived subnetworks of insulin-regulated phosphopeptides.

• Supplementary file 4. Association of kinase enrichment with insulin-stimulated glucose uptake. Pearson's correlation of kinase substrate enrichment analysis (KSEA) enrichment scores with insulin-stimulated glucose uptake for all kinases with Strain or Diet effects. Correlation was performed on all values or on the medians of each strain-diet combination.

• MDAR checklist

### Data availability

All raw and Spectronaut processed phosphoproteomics data have been deposited in the PRIDE proteomeXchange repository and are accessible with the accession PXD041229. Processed data are available as supplementary tables. All code used to analyse data and produce figures has been uploaded to GitHub (copy archived at *van Gerwen, 2023*).

The following dataset was generated:

| Author(s) | Year | Dataset title | Dataset URL | Database and Identifier |
|---|---|---|---|---|
| van Gerwen J, Masson SWC, Cutler HB, Diaz-Vegas A, Potter M, Stöckli J, Madsen S, Nelson ME, Humphrey SJ, James DE | 2023 | The genetic and dietary landscape of the muscle insulin signalling network | https://www.ebi.ac.uk/pride/archive/projects/PXD041229 | PRIDE, PXD041229 |

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
