## [Editor Report · eLife assessment]

This **fundamental** study provides a unique tool for assessing the range of phosphorylation in insulin reactions due to genetic variation and dietary influence through the utilization of genetically distinct mouse strains. The discoveries of this study hold substantial importance, as they shed light on the interplay between genetic attributes and environmental conditions in shaping the insulin-signaling network within skeletal muscle, a crucial regulator of metabolism. The supporting evidence presented is **compelling**, and the work is anticipated to captivate a wide audience within the metabolism discipline due to its extensive appeal and by providing inspiration for further hypothesis-driven research.

---

## [Referee Report · Reviewer #1 (Public Review)]

The authors focused on genetic variability in relation to insulin resistance. The used genetically different lines of mice and exposed them to the same diet. They found that genetic predisposition impacts the overall outcome of metabolic disturbances.

---

## [Referee Report · Reviewer #2 (Public Review)]

Summary:

In the present study, van Gerwen et al. perform deep phosphoproteomics on muscle from saline or insulin-injected mice from 5 distinct strains fed a chow or HF/HS diet. The authors follow these data by defining a variety of intriguing genetic, dietary or gene-by-diet phosphor-sites which respond to insulin accomplished through application of correlation analyses, linear mixed models and a module-based approach (WGCNA). These findings are supported by validation experiments by intersecting results with a previous profile of insulin-responsive sites (Humphrey et al, 2013) and importantly, mechanistic validation of Pfkfb3 where overexpression in L6 myotubes was sufficient to alter fatty acid-induced impairments in insulin-stimulated glucose uptake. To my knowledge, this resource provides the most comprehensive quantification of muscle phospho-proteins which occur as a result of diet in strains of mice where genetic and dietary effects can be quantifiably attributed in an accurate manner. Utilization of this resource is strongly supported by the analyses provided highlighting the complexity of insulin signaling in muscle, exemplified by contrasts to the "classically-used" C57BL6/J strain. As it stands, I view this exceptional resource as comprehensive with compelling strength of evidence behind the mechanism explored. I raised several comments in the last round of assessment but all of them have now been thoughtfully addressed.

Strengths: Generation of a novel resource to explore genetic and dietary interactions influencing the phospho-proteome in muscle. This is accompanied by elegant application of in silico tools to highlight the utility

Weaknesses: none noted

---

## [Referee Report · Reviewer #3 (Public Review)]

Summary:

The authors aimed to investigate how genetic and environmental factors influence the muscle insulin signaling network and its impact on metabolism. They utilized mass spectrometry-based phosphoproteomics to quantify phosphosites in skeletal muscle of genetically distinct mouse strains in different dietary environments, with and without insulin stimulation. The results showed that genetic background and diet both affected insulin signaling, with almost half of the insulin-regulated phosphoproteome being modified by genetic background on an ordinary diet, and high-fat high-sugar feeding affecting insulin signaling in a strain-dependent manner.

Strengths:

Study uses state-of-the-art phosphoproteomics workflow allowing quantification of a large number of phosphosites in skeletal muscle, providing a comprehensive view of the muscle insulin signaling network. The study examined five genetically distinct mouse strains in two dietary environments, allowing for the investigation of the impact of genetic and environmental factors on insulin signaling. The identification of coregulated subnetworks within the insulin signaling pathway expanded our understanding of its organization and provided insights into potential regulatory mechanisms. The study associated diverse signaling responses with insulin-stimulated glucose uptake, uncovering regulators of muscle insulin responsiveness.

Weaknesses:

The limitations acknowledged by the authors, such as the need for larger cohorts and the inclusion of female mice. Moreover as acknowledged by authors, they are unable to dissect to what extent the obesity and different life span cycle for different strain affects insulin signaling. This suggest that further research is needed to validate and expand upon the findings.

---

## [Author Response]

The following is the authors’ response to the original reviews.

**Public Reviews:**

**Reviewer #1 (Public Review):**
The authors focused on genetic variability in relation to insulin resistance. They used genetically different lines of mice and exposed them to the same diet. They found that genetic predisposition impacts the overall outcome of metabolic disturbances. This work provides a fundamental novel view on the role of genetics and insulin resistance.
**Reviewer #2 (Public Review):**
Summary:In the present study, van Gerwen et al. perform deep phosphoproteomics on muscle from saline or insulin-injected mice from 5 distinct strains fed a chow or HF/HS diet. The authors follow these data by defining a variety of intriguing genetic, dietary, or gene-by-diet phosphor-sites that respond to insulin accomplished through the application of correlation analyses, linear mixed models, and a module-based approach (WGCNA). These findings are supported by validation experiments by intersecting results with a previous profile of insulin-responsive sites (Humphrey et al, 2013) and importantly, mechanistic validation of Pfkfb3 where overexpression in L6 myotubes was sufficient to alter fatty acid-induced impairments in insulin-stimulated glucose uptake. To my knowledge, this resource provides the most comprehensive quantification of muscle phospho-proteins which occur as a result of diet in strains of mice where genetic and dietary effects can be quantifiably attributed in an accurate manner. Utilization of this resource is strongly supported by the analyses provided highlighting the complexity of insulin signaling in muscle, exemplified by contrasts to the "classically-used" C57BL6/J strain. As it stands, I view this exceptional resource as comprehensive with compelling strength of evidence behind the mechanism explored. Therefore, most of my comments stem from curiosity about pathways within this resource, many of which are likely well beyond the scope of incorporation in the current manuscript. These include the integration of previous studies investigating these strains for changes in transcriptional or proteomic profiles and intersections with available human phospho-protein data, many of which have been generated by this group.Strengths:Generation of a novel resource to explore genetic and dietary interactions influencing the phospho-proteome in muscle. This is accompanied by the elegant application of in silico tools to highlight the utility.Weaknesses:Some specific aspects of integration with other data among the same fixed strains could be strengthened and/or discussed.
**Reviewer #3 (Public Review):**
Summary:The authors aimed to investigate how genetic and environmental factors influence the muscle insulin signaling network and its impact on metabolism. They utilized mass spectrometry-based phosphoproteomics to quantify phosphosites in the skeletal muscle of genetically distinct mouse strains in different dietary environments, with and without insulin stimulation. The results showed that genetic background and diet both affected insulin signaling, with almost half of the insulin-regulated phosphoproteome being modified by genetic background on an ordinary diet, and high-fat high-sugar feeding affecting insulin signaling in a strain-dependent manner.Strengths:The study uses state-of-the-art phosphoproteomics workflow allowing quantification of a large number of phosphosites in skeletal muscle, providing a comprehensive view of the muscle insulin signaling network. The study examined five genetically distinct mouse strains in two dietary environments, allowing for the investigation of the impact of genetic and environmental factors on insulin signaling. The identification of coregulated subnetworks within the insulin signaling pathway expanded our understanding of its organization and provided insights into potential regulatory mechanisms. The study associated diverse signaling responses with insulin-stimulated glucose uptake, uncovering regulators of muscle insulin responsiveness.Weaknesses:Different mouse strains have huge differences in body weight on normal and high-fat high-sugar diets, which makes comparison between the models challenging. The proteome of muscle across different strains is bound to be different but the changes in protein abundance on phosphosite changes were not assessed. Authors do get around this by calculating 'insulin response' because short insulin treatment should not affect protein abundance. The limitations acknowledged by the authors, such as the need for larger cohorts and the inclusion of female mice, suggest that further research is needed to validate and expand upon the findings.
**Reviewer #1 (Recommendations For The Authors):**
I would suggest further discussion of the potential differences between males and females of the various strains.

In the revised manuscript we have included a more detailed discussion of the potential differences between male and female mice in the "Limitations of this study" section on lines 455-459. In particular, a landmark study of HFD-fed inbred mouse strains found that insulin sensitivity, as inferred from the proxy HOMA-IR, was affected by interactions between sex and strain despite generally being greater in female mice (10.1016/j.cmet.2015.01.002). Furthermore, a recent phosphoproteomics study of human induced pluripotent stem-cell derived myoblasts identified groups of insulin-regulated phosphosites affected by donor sex, and by interactions between sex and donor insulin sensitivity (10.1172/JCI151818). Based on these results, we anticipate that both soleus insulin sensitivity and phoshoproteomic insulin responses would differ between male and female mice through interactions with strain and diet, adding yet another layer of complexity to what we observed in this study. This will be an important avenue for future research to explore.

**Reviewer #2 (Recommendations For The Authors):**
The following are comments to authors - many, if not all are suggestions for extended discussion and beyond the scope of the current elegant study.In the discussion section (line 428) the authors make a key point in that the genetic, dietary, and interacting patterns of variation of Phospho-sites could be due to changes in total protein and/or transcript levels across strains. For example, given the increased expression of Pfkfb3 was sufficient to impact glucose uptake, suggesting that the transcript levels of the gene might also show a similar correlation with insulin responsiveness as in Fig 6b. Undoubtedly, phospho-proteomics analyses will provide unique information on top of more classical omics layers and uncover what would be an important future direction. Therefore, I would suggest adding to the discussion some guidance on performing similar applications to datasets from, at least some, of the strains used where RNA-seq and proteomics are available.

We thank the reviewer for this suggestion. To address this, we mined recently published total proteomics data collected from soleus muscles of seven CHOW or HFD-fed inbred mouse strains, three of which were in common with our study (C57Bl6J, BXH9, BXD34; 10.1016/j.cmet.2021.12.013). In this study ex vivo soleus glucose uptake was measured and correlation analysis was performed, so we directly extracted the resulting glucose uptake-protein associations and compared them to the glucose uptake-phosphoprotein associations identified in our study. Indeed, we found that only a minority of proteins correlated at both the phosphosite and total protein levels, highlighting the utility of phosphoproteomics to provide orthogonal information to more classical omics layers. We have included this analysis in lines 303-311.

Relevant to this, the authors might want to consider depositing scripts to analyze some aspects of the data (ex. WGCNA on P-protein data or insulin-regulated anova) in a repository such as github so that these can be applied easily to other datasets.

We refer the reviewer to the section "Code availability" on lines 511-513, where we deposited all code used to analyse the data on github.

In contrast to the points above, I feel that the short time-course of insulin stimulation was one important aspect of the experimental design that was not emphasized enough as a strength. It was mentioned as a limitation in that other time points could provide more info, yes. But given that the total abundance of proteins and transcripts likely doesn't shift tremendously in this time frame, this provides an important appeal to the analysis of phosphor-proteomic data. I would suggest highlighting the insulin-stimulated response analysis here as something that leverages the unique nature of phosphoproteomics.

We are grateful for the reviewer's positivity regarding this aspect of our experimental design. We have reiterated the value of the 10min insulin stimulation - that it temporally segregates phosphoproteomic and total proteomic changes - in the "Limitations of this study" section on lines 477-481.

While I recognize the WGCNA analysis as an instrumental way to highlight global patterns of phospo-peptide abundance co-regulation, the analysis currently seems somewhat underdeveloped. For example, Fig 5f-h shows a lot of overlap between kinase substrates and pathways among modules. Clearly, there are informative differences based on the intersection with Humphries 2013 and the correlation with Pfkbp3. To highlight the specific membership of these modules, most people rank-order module members by correlation with eigen-gene (or P-peptide) and then perform pathway enrichments on these. Alternatively, it looks like all data was used to generate modules across conditions. One consideration would be to perform WGCNA on relevant comparison data separately (ex. chow mice only and HFHS only) and then compare modules whose membership is retained or shift between the two. Or even look at module representation for genes that show large correlations with insulin-responsiveness. This might also be a good opportunity to suggest readers intersect module members with muscle eQTLs which colocalize to glucose or insulin to prioritize some potential key drivers.

We thank the reviewer for their helpful suggestions, which we feel have substantially improved the WGCNA analysis. To probe specific functional differences between subnetworks, we performed rank-based enrichment using phosphopeptide module membership scores. Interestingly, this did reveal pathways that were enriched only in certain modules. However, we found that after p-value adjustment, virtually all enriched pathways lost statistical significance, hence we interpret these results as suggestive only. We have made this analysis available to readers in Fig S4b-d and lines 263-265: "To further probe functional differences we analysed phosphopeptide subnetwork membership scores, which revealed additional pathways enriched in individual subnetworks. However, these results were not significant after p-value adjustment and hence are suggestive only (Fig. S4b-d)". We also visualised module representation for glucose-uptake correlated phosphopeptides. This agreed with our existing analyis in Fig. 6f, where the eigenpeptides of modules V and I were correlated with glucose uptake (Fig. 6f). We have incorporated this new analysis in Fig. S6b-c and lines 324-325: "Examining the subnetwork membership scores for glucose-uptake correlated phosphopeptides also revealed a preference for clusters V and I, supporting this analysis (Fig. S6b-c)." Finally, in the discussion we have presented the integration of genetic data, such as muscle-specific eQTLs, as a future direction (lines 398-401): "Alternatively, one could overlap subnetworks with genetic information, such as genes associated with glucose homeostasis and other metabolic traits in human GWAS studies, or muscle-specific eQTLs or pQTLs genetically colocalised with similar traits, to further prioritise subnetwork-associated phenotypes and identify potential drivers within subnetworks."

Have the authors considered using their heritability and GxE estimated for module eigenpeptides? To my knowledge, this has never been performed and might provide some informative information as the co-regulated P-protein structure occurs as a result of relevant contexts.

In the revised manuscript we have now analysed eigenpeptides with the same statistical tests used to identify Strain and Diet effects in insulin-regulated phosphopeptides. We have displayed the statistical results in Fig. S4a, and have explicitly mentioned examples of StrainxDiet effects on lines 245-247: "For example, HFD-feeding attenuated the insulin response of subnetwork I in CAST and C57Bl6J strains (t-test adjusted p = 0.0256, 0.0365), while subnetwork II was affected by HFD-feeding only in CAST and NOD (Fig. 5e, Fig. S4a, t-test adjusted p = 0.00258, 0.0256)."

The integration of modules with adipocyte phosphoproteomic data from the authors 2013 Cell metab paper seems like an important way to highlight the integration of this resource to define critical cellular signaling mechanisms. To assess the conservation of signaling mechanisms and relationships to additional key contexts (ex. exercise), the intersection of the insulin-stimulated P-peptides with human datasets generated by this group (ex. cell metab 2015, nature biotech 2022) seems like an obvious future direction to prioritize targets. Figure S3B shows a starting point for these types of integrations.

To demonstrate the value of integrating our results with related phosphoproteomics data, we have incorporated the reviewer's advice of comparing insulin-regulated phosphosites to exercise-regulated phosphosites from Needham et. Nature Biotech 2022 and Hoffman et al. Cell Metabolism 2015. We identified a small subset of commonly regulated phosphosites (8 across all three studies). Given insulin and exercise both promote GLUT4 translocation, these sites may represent conserved regulatory mechanisms. This analysis is presented in Fig. S3d, Table S2, and lines 129-135: "In addition to insulin, exercise also promotes GLUT4 translocation in skeletal muscle. We identified a small subset of phosphosites regulated by insulin in this study that were also regulated by exercise in two separate human phosphoproteomics studies (Fig. S3d, Table S2, phosphosites: Eef2 T57 and T59, Mff S129 and S131, Larp1 S498, Tbc1d4 S324, Svil S300, Gys1 S645), providing a starting point for exploring conserved signalling regulators of GLUT4 translocation."

For the Pfkfb3 overexpression system, are there specific P-peptides that are increased/decreased upon insulin stimulation? This might be an interesting future direction to mention in order to link signaling mechanisms.

We assessed whether canonical insulin signalling was affected by Pfkfb3 overexpression by immunoblotting. Insulin-stimulated phosphorylation of Akt S473, Akt T308, Gsk3a/b S21/S9, and PRAS40 T246 differed little across conditions, with only a weak, statistically insignificant trend towards increased pT308 Akt, pS21/S9 Gsk3a/b, and pT246 PRAS40 in palmitate-treated Pfkfb3-overexpressing cells. Hence, as the reviewer has suggested, an interesting future direction will be to perform phosphoproteomics to characterise more deeply the effects of palmitate and Pfkfb3 overexpression on insulin signalling. We have modified the manuscript to reflect these findings and suggested future directions on lines 362-365: "immunoblotting of canonical insulin-responsive phosphosites on Akt and its substrates GSK3α/β and PRAS40 revealed minimal effect of palmitate treatment and Pfkfb3 overexpression (Fig. S7e-f), hence more detailed phosphoproteomics studies are needed to clarify whether Pfkfb3 overexpression restored insulin action by modulating insulin signalling."

**Reviewer #3 (Recommendations For The Authors):**
This remarkable contribution by the esteemed research group has significantly enriched the field of metabolism. The extensive dataset, intertwined with a sophisticated research design, promises to serve as an invaluable resource for the scientific community. I offer a series of suggestions aimed at potentially elevating the manuscript to an even higher standard.Mouse Weight Variation and Correlation Analysis: The pronounced variances in mouse body weights pose a challenge to meaningful comparisons (Fig S1). Could the disparities in the phosphoproteome between basal and insulin-stimulated conditions be attributed to differences in body weight? Consider performing a correlation analysis. Furthermore, does the phosphoproteome of these mouse strains evolve comparably over time? Do these mice age similarly? Kindly incorporate this information.

We thank the reviewer for the suggested analysis. We found there was a significant correlation between the phosphopeptide insulin response and mouse body weight, either in CHOW-fed mice (Strain effects) or across both diets (Diet effects), for ~ 25% of phosphopeptides that exhibited a Strain or Diet effect. Hence, while there is a clear effect of body weight on insulin signalling, this influences only a small proportion of the entire insulin-responsive phosphoproteome. Notably, insulin was dosed according to mouse lean mass to ensure equivalent dosage received by the soleus muscle, hence any insulin signalling differences associated with body weight are unlikely due to differences in dosing. As the reviewer also alludes to, different strains could have different lifespans. This may result in mice having different biological ages at the time of experimentation, and this in turn could influence insulin signalling. This possibility is challenging to assess in a quantitative manner because lifespan data is not available for most strains used. However, it is worth noting that female CAST mice live 77% as long as C57Bl6J mice (median age of 671 vs 866 (10.1073/pnas.1121113109); data is not available for male mice nor the other three strains), and substantial differences in insulin signalling were observed between these two strains. Ultimately, regardless of whether body weight and/or lifespan altered insulin signalling, such differences would still have arisen solely from the distinct genetic backgrounds and diets of the mice, hence we believe they are meaningful results that should not be dismissed. We have added this analysis to the revised manuscript in the "Limitations of this study" section on lines 471-477: "We were also unable to determine the extent to which signalling changes arose from muscle-intrinsic or extrinsic factors. For instance, body weight varied substantially across mice and correlated significantly with 25% of Strain and Diet-affected phosphopeptides (Fig. S8c), suggesting obesity-related systemic factors likely impact a subset of the muscle insulin signalling network. Furthermore, genetic differences in lifespan could alter the “biological age” of different strains and their phosphoproteomes, though we could not assess this possibility since lifespan data are not available for most strains used. "

Soleus Muscle Data and Bias Considerations: Were measurements taken for lean mass and soleus muscle weight? If so, please present the corresponding data.

Measurements for lean mass and the mass of soleus muscle after grinding have been including in Supplementary Figure S1 (panels c-d)

As outlined in the methods section, the variation in protein yield from the soleus muscle across each strain is substantial. Notably, the distinct peptide input for phospho enrichment introduces biases, given that muscles with lower input may exhibit reduced identification (Fig S2). This bias might also manifest in the PCA plot (S2C). Ideally, adopting a uniform protein/peptide input would have been advantageous. Address this concern and contemplate moving the PCA plot to the main figure. It's prudent to reconsider the sentence stating, "Samples from animals of the same strain and diet were highly correlated and generally clustered together, implying the data are highly reproducible (Fig. S2b-d)," particularly if the input and total IDs were not matched.

The reviewer highlights an important point. As the reviewer comments, it would have been our preference to use the same amount of protein material for all samples. However, as there was a wide range in the mass of the soleus muscle across mouse strains (in particular much lower in CAST mice), it was not appropriate to use the same amount of material for all strains. This is indeed evident in the PCA plot (Figure S2c), whereby samples cluster in the second component (PC2) based on the amount of protein material. However, this clustering is not observed in the hierarchical clustering (Figure S2d), and nor are the number of phosphopeptides quantified in each sample substantially impacted by these differences (Figure S2a) as implied by the reviewer. Indeed, the number of phosphopeptides quantified did not noticeably vary when comparing BXH9/BXD34 to C57Bl6J/NOD despite 32.3% less material used, and there were only 12.4% fewer phosphopeptides (average #13891.56 vs 15851.29) in CAST compared to C57Bl6J/NOD strains, despite 51.8% less material used. To further emphasise the minimal effect that input material had on phosphopeptide quantification, we have additionally plotted the number of phosphopeptides quantified in each sample following the filtering steps we employed prior to statistical analysis of the dataset (i.e. ANOVA). This plot (Author response image 1) shows that there is even less variation in the number of quantified phosphopeptides between strains, with only 9.12% fewer phosphopeptides quantified and filtered on average in CAST compared to C57Bl6J/NOD (average #9026.722 vs 9932.711). From a quantitative perspective, in both the PCA (Principal Component 1) and hierarchical clustering analyses, samples are additionally clustered by individual strains, and in the latter they also cluster generally by diet, implying that biological variation between samples remains the primary variation captured in our data. We have modified the manuscript so that these observations are forefront (lines 103-106): "Furthermore, while different strains clustered by the amount of protein material used in the second component of the PCA (Figure S2c), samples from animals of the same strain and diet were highly correlated and generally clustered together, indicating that our data are highly reproducible". To ensure that readers are aware of our decision to alter protein starting material and its implications, we have moved the description of this from the methods to the results, and we have highlighted the impact on phosphopeptide quantification in CAST mice (lines 99-103): "Due to the range in soleus mass across strains (Fig. S1D) we altered the protein material used for EasyPhos (C57Bl6J and NOD: 755 µg, BXH9 and BXD34: 511 µg, CAST: 364 µg), though phosphopeptide quantification was minimally affected, with only 12.4% fewer phosphopeptides quantified on average in CAST compared to the C57lB6J/NOD (average 13891.56 vs 15851.29 Fig. S2a)."

**Author response image 1. sa4fig1:** Phosphopeptide quantification following filtering. a) The number of phosphopeptides quantified in each sample after filtering prior to statistical analysis.

Phosphosite Quantification Filtering: The quantified phosphosites have been dropped from 23,000 to 10,000. Could you elucidate the criteria employed for filtering and provide a concise explanation in the main text?

We thank the reviewer for drawing this ambiguity to our attention. Before testing for insulin regulation, we performed a filtering step requiring phosphopeptides to be quantified well enough for comparisons across strains and diets. Specifically, phosphopeptides were retained if they were quantified well enough to assess the effect of insulin in more than eight strain-diet combinations (≥ 3 insulin-stimulated values and ≥ 3 unstimulated values in each combination). We have now included this explanation of the filtering in the main text on lines 108-114.

ANOVA Choice Clarification: In Figure 4, there's a transition from one-way ANOVA in B to two-way ANOVA in C. Could you expound on the rationale for selecting these distinct methods?

In panel B, we first focussed on kinase regulation differences between strains in the absence of a dietary perturbation. Hence, we performed one-way ANOVAs only within the CHOW-fed mice. In panel C, we then consider the effect of perturbation with the HFD. We perform two-way ANOVAs, allowing us to identify effects of the HFD that are uniform across strains (Diet main effect) or variable across strains (Strain-by-diet interaction).

Cell Line Selection for Functional Experiments: Could you elucidate the rationale behind opting for L6 cells of rat origin over C2C12 mouse cells for functional experiments?

We acknowledge that C2C12 cells have the benefit of being of mouse origin, which aligns with our mouse-derived phosphoproteomics data. However, they are unsuitable for glucose uptake experiments as they lack an insulin-responsive vesicular compartment even upon GLUT4 overexpression, and undergo spontaneous contraction when differentiated resulting in confounding non-insulin dependent glucose uptake (10.1152/ajpendo.00092.2002, 10.1007/s11626-999-0030-8). In contrast, L6 cells readily express insulin-responsive GLUT4, and cannot contract (doi.org/10.1113/JP281352, 10.1007/s11626-999-0030-8). Therefore they are a superior model for studying insulin-dependent glucose transport. We have added a justification of L6 cells over C2C12 cells in the revised manuscript, on lines 352-354: "While L6 cells are of rat origin, they are preferable to the popular C2C12 mouse cell line since the latter lack an insulin-responsive vesicular compartment and undergo spontaneous contraction, resulting in confounding non-insulin dependent glucose uptake."

It's intriguing that while a phosphosite was modulated on Pfkfb2, functional assays were conducted on a different isoform (Pfkfb3) wherein the phosphosite was not detected.

The correlation between Pfkfb2 S469 phosphorylation and insulin-stimulated glucose uptake suggests that F2,6BP production, and subsequent glycolytic activation, positively regulate insulin responsiveness. There are several ways of testing this: (1) Knock down endogenous Pfkfb2, and re-express either wild-type protein or a S469A phosphomutant. If S469 phosphorylation positively regulates insulin responsiveness, then knockdown should decrease insulin responsiveness and re-expression of wild-type Pfkfb2, but not S469A, should restore it. (2) Induce insulin resistance (e.g. through palmitate treatment), and overexpress phosphomimetic S469D or S469E Pfkfb2 to enhance F2,6BP production. Under our hypothesis, this should reverse insulin resistance. (3) There is some evidence that dual phosphorylation of S469 and S486, another activating phosphosite on Pfkfb2, enhances F2,6BP production through 14-3-3 binding (10.1093/emboj/cdg363). Hence, we may expect that introduction of an R18 sequence into Pfkfb2, which causes constitutive 14-3-3 binding (10.1074/jbc.M603274200), would increase Pfkfb2-driven F2,6BP production, and under our hypothesis this should reverse insulin resistance. (4) The paralog Pfkfb3 lacks Akt regulatory sites and has substantially higher basal activity than Pfkfb2. Thus, overexpression of Pfkfb3 should mimic the effect of phosphorylated Pfkfb2, and hence reverse insulin resistance under our hypothesis. While approaches (1), (2), and (3) directly target Pfkfb2, they have drawbacks. For example, (1) may not work if Pfkfb2 knockdown is compensated for by other Pfkfb isoforms, (2) may not work since D/E phosphomimetics often do not recapitulate the molecular effects of S/T phosphorylation (10.1091/mbc.E12-09-0677), and (3) may not work if S469 phosphorylation does not operate through 14-3-3 binding. Hence we performed (4) as it seemed to be the most robust and cleanest experiment to test our hypothesis. We have revised the manuscript to further clarify the challenges of directly targeting Pfkfb2 and the benefits of targeting Pfkfb3 on lines 342-349: "Since Pfkfb2 requires phosphorylation by Akt to produce F2,6BP substantially, increasing F2,6BP production via Pfkfb2 would require enhanced activating site phosphorylation, which is difficult to achieve in a targeted fashion, or phosphomimetic mutation of activating sites to aspartate/glutamate, which often does not recapitulate the molecular effects of serine/threonine phosphorylation. By contrast, the paralog Pfkfb3 has high basal production rates and lacks an Akt motif at the corresponding phosphosites. We therefore rationalised that overexpressing Pfkfb3 would robustly increase F2,6BP production and enhance glycolysis regardless of insulin stimulation and Akt signalling."

Insulin-Independent Action of Pfkfb3: The functionality of Pfkfb3 unfolds in an insulin-independent manner, yet it restores insulin action (Fig 6h). Could you shed light on the mechanism underpinning this phenomenon? Consider measuring F2,6BP concentrations or assessing kinase activity upon overexpression.

Pfkfb3 overexpression increased the glycolytic capacity of L6 myotubes in the absence of insulin stimulation, as inferred by extracellular acidification rate (Fig. S7c). This is indeed consistent with Pfkfb3 enhancing glycolysis through increased F2,6BP concentration in an insulin-independent manner. To shed light on the mechanism connecting this to insulin action, we performed immunoblotting experiments to assess the kinase activity of Akt, a master regulator of the insulin response. Indeed, this experimental direction has precedent as we previously observed that Pfkfb3 overexpression enhanced insulin-stimulated Akt signalling in HEK293 cells, while small-molecule inhibition of Pfkfb kinase activity reduced Akt signalling in 3T3-L1 adipocytes (10.1074/jbc.M115.658815). However, insulin-stimulated phosphorylation of Akt S473, Akt T308, Gsk3a/b S21/S9, and PRAS40 T246 differed little across conditions, with only a weak, statistically insignificant trend towards increased pT308 Akt, pS21/S9 Gsk3a/b, and pT246 PRAS40 in palmitate-treated Pfkfb3-overexpressing cells. Hence, a more detailed phosphoproteomics study will be needed to assess whether Pfkfb3 restores insulin action by modulating insulin signalling. We have described these immunoblotting experiments in lines 361-365 and Fig. S7e-f. We also discussed potential mechanisms through which Pfkfb3-enhanced glycolysis could connect to insulin action in the discussion (lines 427-434).

Figure 6h Statistical Analysis: For the 2DG uptake in Figure 6h, a conventional two-way ANOVA might be more appropriate than a repeated measures ANOVA.

On reflection, we agree that a conventional ANOVA is more appropriate. Furthermore, for simplicity and conciseness we have decided to analyse and present only insulin-stimulated/unstimulated 2DG uptake fold change values in Figure 6h. We have presented all unstimulated and insulin-stimulated values in Figure S7d.